# Decreasing Entropic Regularization Averaged Gradient for Semi-Discrete Optimal Transport

**Ferdinand Genans**[1*]    **Antoine Godichon-Baggioni**[1]
**François-Xavier Vialard**[2]    **Olivier Wintenberger**[1,3]
Sorbonne Université, CNRS, LPSM[1]
Université Gustave Eiffel, CNRS, LIGM[2]
Wolfgang Pauli Institute[3]
{ferdinand.genans-boiteux, antoine.godichon_baggioni,
olivier.wintenberger}@sorbonne-universite.fr
francois-xavier.vialard@u-pem.fr

## Abstract

Adding entropic regularization to Optimal Transport (OT) problems has become a standard approach for designing efficient and scalable solvers. However, regularization introduces a bias from the true solution. To mitigate this bias while still benefiting from the acceleration provided by regularization, a natural solver would adaptively decrease the regularization as it approaches the solution. Although some algorithms heuristically implement this idea, their theoretical guarantees and the extent of their acceleration compared to using a fixed regularization remain largely open. In the setting of semi-discrete OT, where the source measure is continuous and the target is discrete, we prove that decreasing the regularization can indeed accelerate convergence. To this end, we introduce DRAG: Decreasing (entropic) Regularization Averaged Gradient, a stochastic gradient descent algorithm where the regularization decreases with the number of optimization steps. We provide a theoretical analysis showing that DRAG benefits from decreasing regularization compared to a fixed scheme, achieving an unbiased $\mathcal{O}(1/t)$ sample and iteration complexity for both the OT cost and the potential estimation, and a $\mathcal{O}(1/\sqrt{t})$ rate for the OT map. Our theoretical findings are supported by numerical experiments that validate the effectiveness of DRAG and highlight its practical advantages.

## 1 Introduction

Optimal transport is now a widely used framework to compare probability distributions in different areas of data science such as machine learning [14, 25, 6], computational biology [47], imaging [21, 7], even economics [22] or material sciences [9]. The computational and statistical efficiency of OT solvers is the key to facilitating their use in practical applications. Therefore, both computational methods and the statistical bottleneck in OT, often referred to as the curse of dimensionality, have received significant attention over the past decade [42, 57]. Regularization methods such as Entropic OT (EOT) [16] are popular techniques to mitigate these two issues. It consists of adding an entropic regularization term to the objective function. OT and its entropic regularization apply to different contexts of interest. The most general context is when the two distributions are accessed via samples and one wants to estimate the OT distance and the corresponding plan or map. Another context of interest in some applications, that has recently gained popularity in generative modeling [2, 10, 36], is the case of semi-discrete OT. In this setting, one of the two distributions is discrete and the other continuous. The OT problem, while being a natural proxy to the continuous case, is then slightly simpler since (i) it reduces to the estimation of Laguerre cells and (ii) the curse of dimensionality is alleviated [45].

39th Conference on Neural Information Processing Systems (NeurIPS 2025).

**Related works.** The incorporation of entropic regularization for OT was pioneered in the discrete setting by Cuturi [16], showing that the Sinkhorn algorithm [53] can efficiently solve the EOT problem. Sinkhorn leads to an $\varepsilon$-accurate OT plan in $\mathcal{O}(n^2/\varepsilon^2)$ time when both measures have $n$ points [19]. However, the poor dependence on $\varepsilon$, observed both theoretically and empirically, motivated annealing strategies (also called $\varepsilon$-scaling), which gradually decrease the regularization during optimization to better approximate the true OT solution. Such schemes appear to significantly improve performance in practice [34, 48, 20], but their theoretical analysis remains largely open [52, 48, 12]. From a statistical perspective, it has recently been shown that decreasing the regularization is not only computationally beneficial but also statistically necessary when using EOT: [45] demonstrate in the semi-discrete setting that with $t$ samples from the source and target measures, taking $\varepsilon \asymp 1/\sqrt{t}$ allows one to achieve a minimax-optimal $\mathcal{O}(1/\sqrt{t})$ rate for OT map estimation, thereby escaping the curse of dimensionality without assuming smoothness of the transport map. Their analysis leverages convergence results for entropic potentials [1, 18] and builds upon the entropic map estimator developed in [51, 44].

In parallel, there has been increasing interest in solving semi-discrete OT problems, where the source distribution is continuous while the target measure is known and discrete [2, 54]. In low dimensions, when the source density is fully known, Newton-type solvers [38, 35, 31] have been proposed, offering highly efficient methods for solving the semi-dual problem. In higher dimensions, or when the source measure is only accessible via samples [24] propose solving the semi-dual formulation of semi-discrete (E)OT using an SGD scheme. SGD solvers are a natural choice here since they are well-suited for large-scale applications: they can operate in an online setting using one sample at a time without storage requirements, and they avoid discretization bias. The study of SGD and Averaged SGD (ASGD) for solving the semi-dual of EOT was further investigated in [5], revealing, however, prohibitive constants in terms of $1/\varepsilon$ and higher. This study shows that, as in the discrete case, the use of entropic regularization introduces a computational trade-off.

**Contributions.** We introduce DRAG (Decreasing Regularization Averaged Gradient), an SGD-based algorithm for solving the non-regularized semi-discrete OT. DRAG employs a decreasing entropic regularization scheme decaying with the sample count, aiming to have the best regularization/accuracy trade-off at any time. While matching the computational and memory efficiency of vanilla SGD, DRAG attains superior convergence compared to fixed-regularization methods by exploiting the enhanced properties of the entropic semi-dual without incurring adverse dependencies on the vanishing regularization. Concretely, given $t$ iid samples from the source measure, DRAG achieves rates up to $\mathcal{O}(1/t)$ for both the OT cost and the potential. A key technical result (Lemma 2) underlying DRAG is that, within an $\varepsilon$-ball around the optimum, the entropic semi-dual with regularization $\varepsilon$ satisfies a restricted strong-convexity property independent of $\varepsilon$. DRAG is designed to remain, with high probability, within this $\varepsilon$-ball, even as $\varepsilon$ decreases. Furthermore, by analyzing the difference between the Laguerre cells induced by the true OT potential and our estimate, we establish a $\mathcal{O}(1/\sqrt{t})$ convergence rate for the OT map. Both our theoretical analysis and numerical experiments confirm the benefits of decreasing regularization incorporated in DRAG.

**Notations.** We note $\| \cdot \|$ the Euclidean norm, and for $\mathcal{C} \subset \mathbb{R}^d$, $D_{\mathcal{C}} := \sup\{\|x - y\| : x, y \in \mathcal{C}\}$ denote its diameter. For $a, b \in \mathbb{R}$, $a \vee b := \max\{a, b\}$ and $a \wedge b := \min\{a, b\}$. For $v \in \mathbb{R}^d$, $v_{\min} := \min_{1 \leq j \leq d} v_j$. $\mathbf{1}_d$ and $\mathbf{0}_d$ denote the vectors $(1, \ldots, 1)$ and $(0, \ldots, 0)$ in $\mathbb{R}^d$. $\lambda_{\mathbb{R}^d}$ is the Lebesgue measure in $\mathbb{R}^d$. $\mathcal{P}(\mathbb{R}^d)$ is the set of probabilities in $\mathbb{R}^d$, and for $\rho \in \mathcal{P}(\mathbb{R}^d)$, $\mathrm{Supp}(\rho)$ is its support. $\mathcal{O}(\cdot)$ and $o(\cdot)$ are the usual approximation orders. We use $f \lesssim g$ if there exists a constant $C > 0$ such that $f(\cdot) \leq Cg(\cdot)$. We write $a \asymp b$ if both $a \lesssim b$ and $b \lesssim a$.

## 2 Behind Stochastic Approximation for Optimal Transport

### 2.1 Background on (Entropic) Optimal Transport

Given a source and target probability measures $\mu, \nu \in \mathcal{P}(\mathbb{R}^d)$, a cost function $c : \mathbb{R}^d \times \mathbb{R}^d \to \mathbb{R}^+$ and a regularization parameter $\varepsilon \geq 0$, the Entropic Optimal Transport (EOT) problem is

$$\mathrm{OT}_c^\varepsilon(\mu, \nu) := \min_{\pi \in \Pi(\mu, \nu)} \int_{\mathbb{R}^d \times \mathbb{R}^d} c(x, y)\mathrm{d}\pi(x, y) + \varepsilon \int_{\mathbb{R}^d \times \mathbb{R}^d} \ln\left(\frac{\mathrm{d}\pi}{\mathrm{d}\mu \otimes \nu}(x, y)\right)\mathrm{d}\pi(x, y), \quad (1)$$

where $\Pi(\mu, \nu)$ is the set of joints probability measures on $\mathbb{R}^d \times \mathbb{R}^d$ with marginals $\mu$ and $\nu$. Mild conditions on $\mu, \nu$ and the cost can be made so that this problem is well-posed, see [56]. When $\varepsilon = 0$,

Problem (1) recovers the Kantorovich formulation of OT. In this article, we focus on the quadratic cost $c(x,y) = \frac{1}{2}\|x-y\|^2$, although some of our results can be extended to other costs. Our analysis relies on the semi-dual formulation of the convex problem (1) given by

$$\mathrm{OT}_c^\varepsilon(\mu,\nu) = \min_{f\in C(\mathbb{R}^d)} -\int_{\mathbb{R}^d} f(x)\mathrm{d}\mu(x) - \int f^{c,\varepsilon}(y)\mathrm{d}\nu(y), \tag{2}$$

where for all $y \in \mathbb{R}^d$,

$$f^{c,\varepsilon}(y) := \begin{cases} \min_{x\in\mathbb{R}^d} c(x,y) - f(x) & \text{if} \quad \varepsilon = 0, \\ -\varepsilon\log\left(\int_{\mathbb{R}^d}\exp\left(\frac{f(x)-c(x,y)}{\varepsilon}\right)\mathrm{d}\mu(x)\right) & \text{if} \quad \varepsilon > 0. \end{cases}$$

Under mild conditions on the cost or densities, a positive $\varepsilon$ makes the semi-dual formulation $1/\varepsilon$-smooth [17]. The key property of this semi-dual formulation of (E)OT is to retain more convexity than the standard dual of (1) (see [28, 55]).

**Optimal map and Brenier's theorem.** We consider the quadratic cost, $\varepsilon = 0$ and $\mu,\nu$ having second-order moments. Under the additional assumption that the measure $\mu$ is absolutely continuous, the optimal potential $f^*$, called Kantorovich potential, is (locally) Lipschitz and the map

$$T_{\mu,\nu}(x) := x - \nabla f^*(x) \tag{3}$$

pushes forward $\mu$ onto $\nu$ (see [8]). In addition, $T_{\mu,\nu}$ is the gradient of a convex function. This optimal map has more importance than the OT cost in subfields of machine learning such as generative modeling [29, 36] or domain adaptation [15].

## 2.2 Semi-discrete OT

Semi-discrete (E)OT is when the source measure $\mu$ is absolutely continuous and the target measure $\nu = \sum_{j=1}^M w_j\delta_{y_j}$ is a finite sum of $M \geq 1$ Dirac masses with weights $w_j > 0$. In this case, the semi-dual formulation reduces to a finite-dimensional convex optimization problem on $\mathbb{R}^M$

$$\min_{\mathbf{g}\in\mathbb{R}^M} H_\varepsilon(\mathbf{g}) \overset{\text{def.}}{=} -\int_{\mathbb{R}^d} \mathbf{g}^{c,\varepsilon}(x)\mathrm{d}\mu(x) - \sum_{j=1}^M g_j w_j, \tag{4}$$

where for all $x \in \mathbb{R}^d$, $\mathbf{g}^{c,\varepsilon}(x)$ is a (vectorial) $(c,\varepsilon)$-transform with respect to a vector $\mathbf{g} = (g_1,\ldots,g_M) \in \mathbb{R}^M$, defined by

$$\mathbf{g}^{c,\varepsilon}(x) = \begin{cases} \min_{j\in[\![1,M]\!]}\left[\frac{1}{2}\|x-y_j\|^2 - g_j\right] & \text{if} \quad \varepsilon = 0, \\ -\varepsilon\ln\left(\sum_{j=1}^M\exp\left(\frac{-\frac{1}{2}\|x-y_j\|^2+g_j}{\varepsilon}\right)w_j\right) & \text{if} \quad \varepsilon > 0. \end{cases}$$

The vector $\mathbf{g}$ corresponds to the value of the potential function at the points $y_j$. For notational convenience, we write $H_\varepsilon(\mathbf{g}) = \int_{\mathbb{R}^d} h_\varepsilon(x,\mathbf{g})\mathrm{d}\mu(x)$ with $h_\varepsilon(x,\mathbf{g}) = -\mathbf{g}^{c,\varepsilon}(x) - \sum_{j=1}^M g_j w_j$. For all $\mathbf{g}\in\mathbb{R}^M$ and given $X\sim\mu$, an unbiased estimator of the gradient is given by

$$\nabla_{\mathbf{g}}h_\varepsilon(X,\mathbf{g})_j = -w_j + \chi_j^\varepsilon(X,\mathbf{g}), \qquad 1 \leq j \leq M,$$

where for $(x,\mathbf{g}) \in \mathbb{R}^d \times \mathbb{R}^M$, we have

$$\chi_j^\varepsilon(x,\mathbf{g}) = \frac{\exp\left(\frac{-\frac{1}{2}\|x-y_j\|^2+g_j}{\varepsilon}\right)w_j}{\sum_{k=1}^M\exp\left(\frac{-\frac{1}{2}\|x-y_k\|^2+g_k}{\varepsilon}\right)w_k}.$$

For $\varepsilon = 0$, $\chi_j(x,\mathbf{g}) = \mathbb{1}_{\mathbb{L}_j(\mathbf{g})}(x)$ is an indicator function and we have a partition $\mathbb{R}^d = \bigcup_{j=1}^M \mathbb{L}_j(\mathbf{g})$, where for all $j \in [\![1,M]\!]$,

$$\mathbb{L}_j(\mathbf{g}) := \left\{x \in \mathbb{R}^d; \mathbf{g}^c(x) = \frac{1}{2}\|x-y_j\|^2 - g_j\right\}.$$

The convex sets $\mathbb{L}_j(\mathbf{g})$ are called power or Laguerre cells and $\mu(\mathbb{L}_i(\mathbf{g})\cap\mathbb{L}_j(\mathbf{g})) = 0$ when $i \neq j$. By the first-order optimality condition, solving semi-discrete OT amounts to finding $\mathbf{g}$ such that for all $j \in [\![1,M]\!]$, $\mu(\mathbb{L}_j(\mathbf{g})) = w_j$. Semi-discrete OT is a case of application of Brenier's theorem. Given the optimal potential $\mathbf{g}^*$, the OT map is $T_{\mu,\nu}(x) = x - \nabla(\mathbf{g}^*)^c(x) = y_j$ for $x$ inside $\mathbb{L}_j(\mathbf{g}^*)$.

## 2.3 Solving semi-discrete (E)OT with the semi-dual formulation

Exploiting its finite-dimensional nature, solving semi-discrete OT by optimizing its semi-dual formulation has become a popular approach. Notably, Newton methods are highly effective to solve $H_0$ in scenarios with low dimensions and known source densities, utilizing meshes to approximate the source density [38, 35, 31]. In scenarios involving arbitrary dimensions or when only sample-based access to the source measure is available, EOT emerges as a favored strategy. Notably, to avoid working with a discretized version of the source measure, such as with the Sinkhorn Algorithm, [24] recommend employing stochastic optimization to solve (4). Indeed, the semi-dual EOT problem has a convex objective of the form

$$H_\varepsilon(\mathbf{g}) = \mathbb{E}_{X \sim \mu}[h_\varepsilon(X, \mathbf{g})],$$

with $X$ as a random variable under $\mu$. As noted in [24], the main advantage of stochastic optimization algorithms is that they are suited for really large-scale problems, keeping in memory only the discrete measure $\nu$. Moreover, avoiding discretization enables unbiased estimation of (E)OT quantities. SGD-based solvers also naturally operate in an online fashion, progressively refining their solutions as more samples become available.

For a given fixed regularization parameter $\varepsilon > 0$, stochastic first-order methods are predominantly employed to solve (4). Starting with an initial value $\mathbf{g}_0 \in \mathbb{R}^M$, these algorithms consider at each iteration one or many samples $X_t \sim \mu$ and rely on an update of the form

$$\mathbf{g}_t = \mathbf{g}_{t-1} - \gamma_t \nabla_{\mathbf{g}} h_\varepsilon(X_t, \mathbf{g}_{t-1}).$$

At time $t$, the Averaged Stochastic Gradient Descent (ASGD) returns the averaged estimate $\overline{\mathbf{g}}_t = \frac{1}{t+1} \sum_{k=0}^{t} \mathbf{g}_k$, while Stochastic Gradient Descent (SGD) returns $\mathbf{g}_t$. ASGD, as an acceleration of SGD, has been widely studied in the literature (see [43, 41, 4], and [5] for the specific case of EOT).

**Choosing the regularization parameter $\varepsilon$ for EOT.** Approximating the EOT problem rather than the OT one benefits from an enhanced convergence rate, especially in the discrete setting. The introduction of the Sinkhorn Algorithm for solving the EOT problem, as highlighted by [16], has led to a resurgence of interest in OT within the Machine Learning community.

The choice of the regularization parameter $\varepsilon$ then becomes a practical and/or statistical problem:

1. Selecting the regularization parameter is a practical issue that aims to strike an optimal balance between convergence speed and accuracy [16, 19]. To address this trade-off, some heuristics, such as $\varepsilon$-scaling [49], which involves a decreasing regularization scheme, are employed in the discrete setting, although they lack sharp theoretical guarantees.

2. In the semi-discrete and continuous settings, the initial statistical problem is to determine the number of samples needed to accurately approximate the OT quantities. In this line of work, the use of EOT to construct estimators has also been proven to be satisfactory. In this case, studies show that regularization must decrease as the number of samples increases [44, 45]. However, discrete solvers do not adjust to the number of drawn points, as the solver is initiated once the points to approximate the measures have been sampled.

## 3 DRAG: Decreasing Regularization Averaged Gradient

### 3.1 The setting.

We focus here on the one-sample setting of semi-discrete OT. Specifically, we sample from the source measure $\mu$ and leverage the full information of the discrete measure $\nu$. Furthermore, fixing $R > 0$ and $\alpha \in (0, 1]$, we make the following mild assumption, already present in [18, 45].

**Assumption 1.** *We assume that $\mu \in \mathcal{P}(\mathbb{R}^d)$ has support contained in the convex ball $B(0, R)$ and admits a density $d\mu$ that is $\alpha$-Hölder continuous and satisfies $0 < d\mu < \infty$ on its support. We denote by $\mathcal{P}_\alpha(B(0, R))$ the set of such measures.*
*The target $\nu$ is assumed to be discrete, of the form $\nu = \frac{1}{M} \sum_{j=1}^{M} \delta_{y_j}$, with $(y_1, \ldots, y_M) \in B(0, R)^M$.*

## 3.2 DRAG: A gradient-based algorithm adaptive to both the sample size and the regularization parameter

To accurately estimate the non-regularized OT cost and map, it is crucial to use a regularization parameter $\varepsilon$ that decreases as the number of drawn samples increases. However, no existing algorithm in the OT literature simultaneously adapts to both entropic regularization and sample size. Inspired by $\varepsilon$-annealing [49], a decreasing regularization scheme from the discrete OT setting, which is known for accelerating the convergence of the Sinkhorn algorithm in practice, and considering that SGD algorithms are inherently adaptive to the number of samples, we introduce the Decreasing entropic Regularization projected Averaged stochastic Gradient descent (DRAG) to solve the semi-dual (2). Our algorithm employs a decreasing regularization sequence $(\varepsilon_t)_t$ and replaces the usual gradient step in SGD with a projected step using adaptive regularization

$$\mathbf{g}_t = \mathrm{Proj}_{\mathcal{C}} \left( \mathbf{g}_{t-1} - \gamma_t \nabla_{\mathbf{g}} h_{\varepsilon_{t-1}}(X_t, \mathbf{g}_{t-1}) \right),$$

where for $U \subset \mathbb{R}^M$ convex, we define the projector as $\mathrm{Proj}_U(\mathbf{g}) := \arg\min\{\|\mathbf{g} - \mathbf{g}'\|, \mathbf{g}' \in U\}$. This method can be interpreted as a decreasing bias SGD scheme. For such a method, employing a projection step can be highly effective in ensuring convergence [13, 23]. In the context of EOT with bounded cost, it is well established that the $(c, \varepsilon)$-transform enables the localization of a $\|.\|_\infty$-ball, where a minimum of the semi-dual problem lies [40]. Specifically, since $\sup\{c(x, y_j); x \in \mathrm{Supp}(\mu), j \in [\![1, M]\!]\} < 2R^2$ by Assumption 1, a preliminary projection set can be expressed as $\mathcal{C}_\infty := [0, 2R^2]^M$ and we know that we can search for a minimum in this set. Nonetheless, leveraging the regularity of the cost function, we can have a projection set with a unique optimizer, as described in the following Lemma.

**Lemma 1.** *(Proof in Appendix B.6) Under Assumption 1, for all $\varepsilon \geq 0$, there exists a unique solution $\mathbf{g}_\varepsilon^*$ to (4) in $\mathcal{C}_u := \{\mathbf{g} \in \mathbb{R}^M ; g_1 = 0 \text{ and } |g_j| \leq R\|y_1 - y_j\|, j \in [\![1, M]\!]\}$.*

Note that the choice $g_1 = 0$ is arbitrary. In what follows, we refer to $\mathcal{C} = \mathcal{C}_\infty$ or $\mathcal{C} = \mathcal{C}_u$ as our projection set. Note that for both sets, the projection's computational complexity is only $\mathcal{O}(M)$, as it involves merely clipping each coordinate of our vector.

Finally, we consider the Decreasing Regularization projected Averaged stochastic Gradient descent (DRAG) defined by

$$\overline{\mathbf{g}}_t = \frac{1}{t+1}\mathbf{g}_t + \frac{t}{t+1}\overline{\mathbf{g}}_{t-1},$$

with $\overline{\mathbf{g}}_0 = \mathbf{g}_0$. The pseudo-code of our algorithm is given in Algorithm 1. A main advantage of DRAG is its $\mathcal{O}(dtM)$ computational complexity and $\mathcal{O}(dM)$ spatial complexity, which make it well suited for large-scale problems.

---

**Algorithm 1** DRAG

**Parameters:** $(\gamma_1, a, b, \mathcal{C})$
Initialize $\mathbf{g}_0 \in \mathcal{C}$, $\overline{\mathbf{g}}_0 = \mathbf{g}_0$, $\varepsilon_0 = 1$.
**for** $k = 1$ to $t$ **do**
    $\gamma_k = \gamma_1 k^{-b}$
    $X_k \sim \mu$
    $\mathbf{g}_k = \mathrm{Proj}_{\mathcal{C}} \left( \mathbf{g}_{k-1} - \gamma_k \nabla_{\mathbf{g}} h_{\varepsilon_{k-1}}(X_k, \mathbf{g}_{k-1}) \right)$
    $\overline{\mathbf{g}}_k = \frac{1}{k+1}\mathbf{g}_k + \frac{k}{k+1}\overline{\mathbf{g}}_{k-1}$
    $\varepsilon_k = k^{-a}$
**end for**
**return** $\overline{\mathbf{g}}_t$

---

## 3.3 Key properties of the semi-dual $H_\varepsilon$ for the design of DRAG

The design and convergence analysis of DRAG rely on two fundamental properties of the semi-dual objective $H_\varepsilon$. First, the fast convergence of entropic potentials ensures that the optimal solution does not change abruptly as the regularization parameter varies. Second, the enhanced Restricted Strong Convexity (RSC) of $H_\varepsilon$ around its optimum. These two properties are crucial to the construction of DRAG and are detailed below.

**Convergence of the entropic potentials.** The following result from [18] establishes that the convergence of entropic optimal potentials is faster than linear as $\varepsilon' \to \varepsilon$. Note that this result is only given for the quadratic cost and is, therefore, the limiting factor in our analysis for broadening the class of cost functions for DRAG.

**Proposition 1.** *[Corollary 2.2 [18]] For $0 \leq \varepsilon' \leq \varepsilon$, under Assumption 1 , there exists a constant $K_0$, notably depending on the characteristics of $\nu$, such that for any $\alpha' \in (0, \alpha)$,*

$$\|\mathbf{g}_\varepsilon^* - \mathbf{g}_{\varepsilon'}^*\| \leq K_0 \varepsilon^{\alpha'} (\varepsilon - \varepsilon').$$

The constant $K_0$ can depend on the source measure $\mu$, through the radius $R$ and the constants $m_1, m_x 2$ such that $m_1 < d\mu < m_2$ on its support, see Remark 2.1 in [18]. However, terms depending on $K_0$ will be asymptotically negligible in the analysis of DRAG.

Note that, the semi-dual $H$ has the invariance $H(\mathbf{g}) = H(\mathbf{g} + a\mathbf{1}_M)$ for any $\mathbf{g} \in \mathbb{R}^M$ and $a \in \mathbb{R}$. Therefore, the minimizer $\mathbf{g}^*$ of the semi-dual is unique only up to a transformation of the form $\mathbf{g}^*_{\varepsilon_t} + a\mathbf{1}_M$, where $a \in \mathbb{R}^*$. Consequently, our analysis on the orthogonal complement of the subspace spanned by $\mathbf{1}_M$, denoted as $\text{Vect}(\mathbf{1}_M)^\perp$. For simplicity, for $\mathbf{g}, \mathbf{g}' \in \mathbb{R}^M$, we denote for $p \in [1, \infty]$

$$\|\mathbf{g} - \mathbf{g}'\|_p := \|\mathbf{g} - \mathbf{g}'\|_{p\, \text{Vect}(\mathbf{1}_M)^\perp}, \qquad \langle \mathbf{g}, \mathbf{g}' \rangle := \langle \mathbf{g}, \mathbf{g}' \rangle_{\text{Vect}(\mathbf{1}_M)^\perp}.$$

**Global and local Restricted Strong Convexity.** The convergence behavior of gradient-based methods is a central topic in convex optimization. The Restricted Strong Convexity (RSC) condition [58] offers a strictly weaker alternative to strong convexity while still providing comparable guarantees in many settings [58, 50]. The following lemma characterizes the RSC of $H_\varepsilon$. Notably, while the global RSC constant on $\mathcal{C}$ scales linearly with $\varepsilon^{-1}$, the local RSC in a neighborhood of radius $\varepsilon/2$ around the optimum becomes independent of $\varepsilon$. This motivates the decreasing regularization scheme in DRAG: by gradually reducing $\varepsilon$, we ensure that iterates remain within regions where the improved convexity properties can be fully exploited.

**Lemma 2** (Global and local RSC of $H_\varepsilon$, proof in Appendix B.7)**.** *For any $\varepsilon \in (0, 1]$, under Assumption 1, there exists $\rho_* > 0$ independant of $\varepsilon$, such that for all $\mathbf{g} \in \mathcal{C}$,*

$$\langle \nabla H_\varepsilon(\mathbf{g}), \mathbf{g} - \mathbf{g}^*_\varepsilon \rangle \geq \begin{cases} \rho_* \dfrac{\varepsilon}{2c_\infty} \left(1 - e^{-\frac{2c_\infty}{\varepsilon}}\right) \|\mathbf{g} - \mathbf{g}^*_\varepsilon\|^2 & \text{if } \|\mathbf{g} - \mathbf{g}^*_\varepsilon\| > \dfrac{\varepsilon}{2}, \\ \rho_* \left(1 - e^{-1}\right) \|\mathbf{g} - \mathbf{g}^*_\varepsilon\|^2 & \text{if } \|\mathbf{g} - \mathbf{g}^*_\varepsilon\| \leq \dfrac{\varepsilon}{2}. \end{cases}$$

Here, $\rho_*$ provides a lower bound on the strong convexity constant of $H_\varepsilon$ restricted to the subspace $\text{Vect}(\mathbf{1}^\perp)$, and it holds uniformly over $\varepsilon \in (0, 1]$ (see Theorem 3.2 in [18] for further details).

### 3.4 Convergence rate of DRAG

**Convergence rate before averaging.** The following proposition provides a key high-probability control, ensuring that the iterates $\mathbf{g}_t$ remain uniformly close to the optimal potential $\mathbf{g}^*_{\varepsilon_t}$ at all times $t$.

**Proposition 2.** *(Proof in Appendix B.5) Under Assumption 1 with $\mu \in \mathcal{P}_\alpha(B(0, R))$, taking the parameters $(\gamma_1, a, b)$ of DRAG such that $\gamma_1 > 0, b \in \left(\frac{1}{2}, 1\right)$, with constraints $2a < b, a + b < 1, 1 + a + a\alpha > 2b$, we have for any $\delta > 0$ and every $q > 0$,*

$$\mathbb{P}\left(\|\mathbf{g}_t - \mathbf{g}^*_{\varepsilon_t}\| \geq \varepsilon_t\right) \lesssim t^{-q(b-2a)+\delta}.$$

This result is key to leveraging the locally enhanced RSC of $H_{\varepsilon_t}$ and guides how quickly the regularization can decay. When $b > 2a$, it yields a convergence rate of $o(t^p)$ for all $p$. This proposition leads to the convergence rate of the non-averaged DRAG iterates stated in Theorem 1.

**Dependence on $a$, $b$, and $\alpha$.** As we can see, the convergence rate depends on $a, b$ from DRAG and the Hölder regularity. While the constraints may seem difficult to interpret, setting $a$ arbitrarily close to $\frac{1}{3}$ (denoted $a = \frac{1}{3}^-$) and $b = \frac{2}{3}$ ensures that the constraints are satisfied for any $\alpha \in (1/2, 1]$ and the best converge rate for DRAG in our convergence analysis.

**Theorem 1.** *(Proof in Appendix B.1) Under the same assumptions as in Proposition 2, we have for any $\alpha' \in (0, \alpha)$*

$$\mathbb{E}\left[\|\mathbf{g}_t - \mathbf{g}^*\|^2\right] \lesssim \frac{1}{\rho_* \cdot t^b} + \frac{1}{t^{2a+2a\alpha'}}, \qquad t \geq 1.$$

Remarkably, we achieve a convergence rate without any undesirable dependence on regularization. In contrast, [5] derived a convergence rate of the form $\mathcal{O}(\varepsilon^{-c} t^{-b})$ for a fixed regularization, with $c$ at least equal to $1$. Note that having no adverse dependence on the regularization parameter is a key necessary characteristic of DRAG, which aims to solve the non-regularized OT problem at fast rates. This is indeed the case here, and there is no trade-off with the choice of the admissible $a$ and $b$ since the result holds for all $q > 0$.

**Enhanced convergence rate with averaging.** In convex stochastic optimization, it is known that averaging SGD iterations can lead to acceleration. More precisely, ASGD can adapt to the possibly unknown local strong convexity of the objective function at the optimizer [43, 3] and achieve optimal $\mathcal{O}(1/t)$ converge rates. Despite the fact that our objective function changes at each time $t$, Theorem 2 shows that DRAG fully exploits the acceleration thanks to averaging.

**Theorem 2.** *(Proof in Appendix B.2) Under the same assumptions as in Theorem 1, noting $s = \min\{1, 2a + 2a\alpha'\}$, we have for all $\alpha' \in (0, \alpha)$,*

$$\mathbb{E}\left[\|\overline{\mathbf{g}}_t - \mathbf{g}^*\|^2\right] \lesssim \frac{1}{t^s} .$$

The rates again depend on $a$, $b$, and $\alpha$. The key message here is that by taking $a = \frac{1}{3}^-$ and $b = \frac{2}{3}$, we recover the optimal $\mathcal{O}(1/t)$ rate for any $\alpha \in (\frac{1}{2}, 1]$.

# 4 Optimal Transport cost and map estimation with DRAG

In the previous section, we established the convergence rate of DRAG to the OT potential. While this result was central to our theoretical analysis, our final objective is to estimate the OT cost and transport map. Leveraging the convergence rate of the potential, we derive estimation guarantees for these key OT quantities.

## 4.1 OT cost estimation

**Corollary 1.** *(Proof in Appendix B.4) Taking the same assumptions as Theorem 2, with $s = \min\{1, 2a + 2a\alpha'\}$, we have*

$$\mathbb{E}\left|H_0(\mathbf{g}^*) - H_0(\overline{\mathbf{g}}_t)\right| \lesssim \frac{1}{t^s} . \tag{5}$$

Once again, when $\alpha > 1/2$ and setting $a = \frac{1}{3}^-$ and $b = \frac{2}{3}$, we achieve an $\mathcal{O}(1/t)$ convergence rate, which is optimal for strongly convex objectives. This rate is obtained by leveraging the locally enhanced RSC property and our adaptive decreasing regularization scheme. The fact that $H_0$ is locally smooth, as noted in Theorem 4.1 of [32], also plays a crucial role in the proof of Corollary 2 .

## 4.2 Brenier map estimation

When employing entropic regularization, a popular choice to approximate the OT map involves using the estimator of the entropic Brenier map [44] $T_{\mu,\nu}^\varepsilon(\mathbf{g}_\varepsilon^*)(x) = x - \nabla(\mathbf{g}_\varepsilon^*)^{c,\varepsilon}$. Indeed, for $\hat{\mathbf{g}} \in \mathbb{R}^M$, $T_{\mu,\nu}^\varepsilon(\hat{\mathbf{g}})(x)$ could serve as an estimator. The objective is then to find an accurate estimator, $\hat{\mathbf{g}}$, close to $\mathbf{g}_\varepsilon^*$, and to analyze its performance based on the bias-variance decomposition

$$\|T_{\mu,\nu} - T_{\mu,\nu}^\varepsilon(\hat{\mathbf{g}})\|_{L^2(\mu)}^2 \lesssim \|T_{\mu,\nu}^\varepsilon(\hat{\mathbf{g}}) - T_{\mu,\nu}^\varepsilon(\mathbf{g}_\varepsilon^*)\|_{L^2(\mu)}^2 + \varepsilon,$$

using the fact that $\|T_{\mu,\nu} - T_{\mu,\nu}^\varepsilon(\mathbf{g}_\varepsilon^*)\|_{L^2(\mu)}^2 \lesssim \varepsilon$ ([45], Theorem 3.4). However, the mapping $\mathbf{g} \mapsto T_{\mu,\nu}^\varepsilon(\mathbf{g})$ is $\varepsilon^{-1}$-Lipschitz, complicating the bias-variance trade-off given that $\varepsilon_t = t^{-a}$. Instead, we rely on the gradient computed thanks to the $c$-transform of the estimator $\overline{\mathbf{g}}_t$ of DRAG. In fact, for any $x \in \mathbb{R}^d$, if there exists $j \in [\![1, M]\!]$ such that $x$ is in the interior of $\mathbb{L}_j(\mathbf{g}^*) \cap \mathbb{L}_j(\overline{\mathbf{g}}_t)$, we have $T_{\mu,\nu}(x) = x - \nabla(\overline{\mathbf{g}}_t)^c(x)$. Indeed, no matter $\mathbf{g}$, whenever $x \in \mathbb{R}^d$ is in the interior of $\mathbb{L}_j(\mathbf{g})$, the gradient of $\mathbf{g}^c$ is given by

$$\nabla(\mathbf{g})^c(x) = \arg\max_k \left[\frac{1}{2}\|x - y_k\|^2 - g_j\right] = y_j. \tag{6}$$

By analyzing the differences of Laguerre cells partitions between $\mathbb{L}(\overline{\mathbf{g}}_t)$ and $\mathbb{L}(\mathbf{g}^*)$, we derive the following theorem.

**Theorem 3.** *(Proof in Appendix B.3) Under the same assumptions as Theorem 1, defining for all $x \in \mathbb{R}^d$ and time $t \geq 0$ $T(\overline{\mathbf{g}}_t)(x) = x - \nabla\overline{\mathbf{g}}_t^c$, $s = \min\{1, 2a + 2a\alpha'\}$, we have for all $1 \leq p < \infty$*

$$\mathbb{E}\left[\|T_{\mu,\nu} - T_{\mu,\nu}(\overline{\mathbf{g}}_t)\|_{L^p(\mu)}^p\right] \lesssim \frac{1}{t^{s/2}} .$$

When $\alpha > 1/2$, setting $a = \frac{1}{3}^-$ and $b = \frac{2}{3}$, we recover an $\mathcal{O}(1/\sqrt{t})$ convergence rate. This matches the rate obtained in [45], but in our case it is achieved in the one-sample setting with an algorithm that refines its estimate online, whereas their approach relies on the Sinkhorn algorithm in a batched setting. Note that, unlike our method, theirs achieves the optimal rate for any $\alpha \in (0,1)$. However, the use of online algorithms is crucial in high-dimensional applications where data is sampled sequentially, such as in generative modeling. In such settings, although the source measure is not compact, it is often chosen to be the standard Gaussian, which has a Lipschitz density and therefore corresponds to the case $\alpha = 1$.

## 5 Numerical experiments

**Convergence rates of DRAG on synthetic data.** We numerically verify here our convergence rate guarantees through various examples. For each example, we know the theoretical OT map, cost, and discrete potential. The first two examples are similar to those in [45]. In all figures and experiments, we set the parameters of DRAG to $\varepsilon_t = 0.1/t^a$, $a = 0.33$, $b = 2/3$, $\gamma_1 = \text{Diam}(\text{Supp}(\mu))$. Our numerical investigation found that our parameter selection is robust without further hypertuning.

**Examples settings: (1)** $\mu \sim \mathcal{U}([0,1]^{10^3})$, $\text{Supp}(\nu) = \{y_j = (\frac{j-1/2}{M}, \frac{1}{2}, ..., \frac{1}{2}), j \in [\![1, M]\!]\}$, $\mathbf{w} = \frac{1}{M}\mathbf{1}_M$, $M = 1000$. **(2)** $\mu \sim \mathcal{U}([0,1]^{10^3})$, $M = 30$ and $y_1, ..., y_M$ randomly generated in $[0,1]^{10^3}$. We then also randomly generate $\mathbf{g}^* \in \mathbb{R}^{30}$ and approximate $\mathbf{w}$ with Monte Carlo (MC), such that $g^*$ is the discrete optimum potential. This setting led to non-uniform weights, with $w_{\min} = 0.001$. **(3)** $\mu \sim \mathcal{U}([\delta, 1 + \delta])$, $\delta = 0.5$, $\text{Supp}(\nu) = \left\{\frac{k}{M}; k \in [\![1, M]\!]\right\}$, $\mathbf{w} = \frac{1}{M}\mathbf{1}_M$, $M = 1000$.

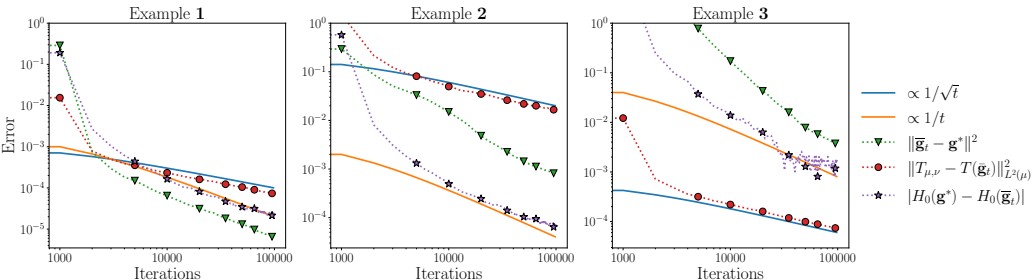

Figure 1: Convergence rate to the OT potential, cost and map for Examples **1**,**2** and **3**.

In Figure 1, we show the convergence rates of the OT cost, map, and discrete potential. As we can see, our theoretical rates are matched for all OT quantities. The higher variance in the OT cost estimations in Example **3** is likely due to the use of $10^8$ Monte Carlo samples to approximate $H$, which introduces an additional approximation error beyond the one caused by DRAG alone.

**Dependence on the dimension.** For DRAG, the ambient dimension of the measures does not change the convergence rate but the number of points $M$ does. This is due to the fact that, with the semi-dual formulation, we solve an expected minimization problem, and the dimension is then the number of points of our discrete measure. For instance, changing the dimension from $d = 10$ to $d = 1000$ does not change the convergence rate. However, if one wants to approximate continuous OT with semi-discrete OT and DRAG, the accuracy of the approximation of the continuous measure with $M$ points will depend on the dimension and thus will have an impact on DRAG.

**DRAG compared to fixed regularization ASGD.** In Figures 2 and 3, we compare the effectiveness and robustness of decreasing vs. fixed regularization schemes. Figure 2 shows that DRAG consistently outperforms projected ASGD with various fixed regularization values, achieving a better trade-off between convergence speed and solution quality. Fixed schemes either converge to biased solutions when regularization is large or fail to converge in time when it is too small (e.g., $\varepsilon = 5 \cdot 10^{-3}$). This highlights DRAG's advantage during the entire optimization process and supports the idea that starting with high regularization and gradually reducing it yields more stable and accurate solutions in semi-discrete optimal transport. Figure 3 shows DRAG's robustness to the decay parameter $a$: both $a = 0.3$ and $a = 0.5$ yield similar convergence, indicating low sensitivity. All decreasing regularization variants also clearly outperform the non-regularized projected ASGD. While all

regularization schemes eventually converge with more iterations, DRAG remains one to two orders of magnitude more accurate, due to its improved start visible in both figures (see Appendix A).

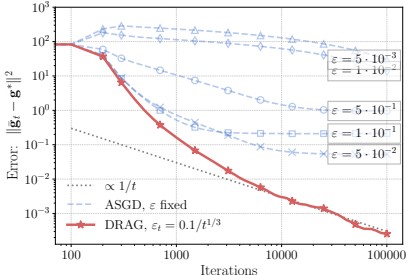

Figure 2: DRAG compared to ASGD with a fixed regularization on Example **1**.

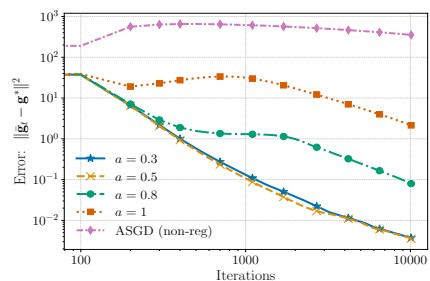

Figure 3: DRAG, with different decreasing regularization rate $\varepsilon_t = 0.1/t^a$ on Example **1**.

**Generative modeling task.** We illustrate the practical benefits of our solver in the context of generative modeling. In [2], semi-discrete OT is used to map a simple prior onto encoded data points in latent space, with the goal of reducing mode collapse. To generate new samples, they approximate a semi-discrete OT map from a standard gaussian to the empirical distribution of encoded data points in the latent space and then apply a specific interpolation scheme to obtain a continuous mapping from prior to latent space. We replicate their pipeline on toy datasets from their repository [27], replacing their ADAM-based solver with DRAG, using the same number of samples. As we can see, while both solvers yield good results on the "swissroll" target data, DRAG outperforms the ADAM solver on the "spiralarms" data, being able to almost completely generate it, whereas ADAM shows poorer coverage. Since the Gaussian is not compactly supported, this setup falls under Assumption 1, and DRAG was run without projections. This further underscores its robustness in more general settings.

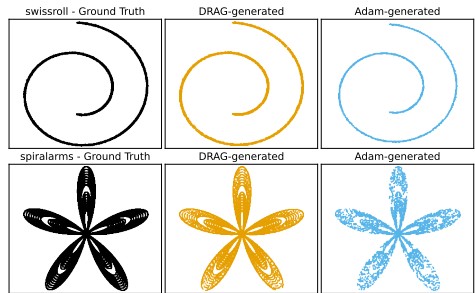

Figure 4: Comparison of DRAG and ADAM for a generative model task

**Monge-Kantorovich Quantiles.** We visualize our OT map estimator on a concrete example of Monge-Kantorovich (MK) quantiles [11]. In this context, having a target measure $\nu$ to investigate, the source measure is set to be the uniform measure on the unit Euclidean ball $\mu \sim \mathcal{U}(B(0,1))$. Given the OT map $T_{\mu,\nu}$ (or its approximation), $T_{\mu,\nu}\big(B(0,k/10)\big)$ for $k \in [\![1, 10]\!]$ define MK quantile regions. We used $M = 10^5$ points to approximate $\nu$, a discrete version of a boomerang-shaped measure and benchmarked DRAG against two OT solvers that can solve semi-discrete OT: Online Sinkhorn [37], using the EOT map estimator and Neural OT [33]. DRAG and Online Sinkhorn used $10^7$ source samples; for the latter, the entropic regularisation was tuned to $\varepsilon = 10^{-3}$. Both ran in under one minute. Neural OT, following Appendix B of [33], processed over $10^8$ samples using a three hidden layers MLP, was ten times slower, even on a A100 GPU. Figure 5 displays the estimated MK quantile regions of the target measure $\nu$, color-coding each centered annulus region. As visible in the figure, DRAG is the only method producing an unbiased estimate of the MK quantiles, fully covering the support of $\nu$ while keeping every MK region convex, as expected in theory.

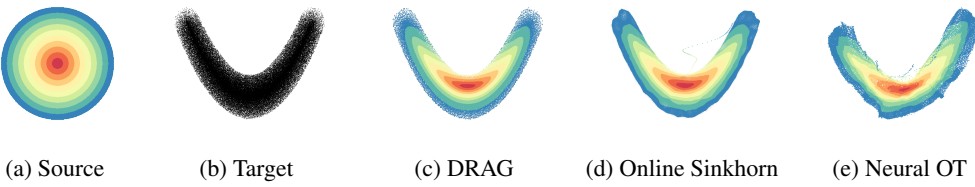

(a) Source     (b) Target     (c) DRAG     (d) Online Sinkhorn     (e) Neural OT

Figure 5: Comparison of Monge-Kantorovich quantiles approximation with different solvers

**Additional experiments.** The appendix presents further experiments that, while not affecting our theoretical results, may benefit practitioners. We show that mini-batching with GPU acceleration and weighted averaging of iterates $\mathbf{g}_t$ can significantly speed up the algorithm. We also compare DRAG to Adam on synthetic data, highlighting its superior performance.

## 6 Conclusion

In EOT, a decreasing regularization parameter naturally appeals to practitioners who employ annealing schemes to accelerate Sinkhorn-like algorithms. Similarly, in the statistical community, regularization that decreases with the number of samples is preferred for more accurately approximating true OT quantities. With our algorithm, DRAG, we show that these two motivations for decreasing regularization can successfully coexist by proving that a decreasing entropic regularization scheme can indeed improve the convergence rate, whereas it was only a heuristic beforehand. Moreover, we prove that DRAG achieves optimal convergence rates: $\mathcal{O}(1/t)$ for both the OT potential and cost, and $\mathcal{O}(1/\sqrt{t})$ for the OT map. These rates are obtained by leveraging decreasing regularization as a form of acceleration. To the best of our knowledge, this is the first algorithm in the OT literature that adapts to both regularization strength and sample size. Our results also motivate further investigation of decreasing regularization in (i) discrete OT, by adapting our approach to demonstrate the acceleration benefits of annealing schemes, and in (ii) semi-discrete OT, by developing new optimized versions of DRAG, such as those incorporating adaptive step sizes in a similar way to Adam or Adagrad.

## 7 Acknowledgements

The work of François-Xavier Vialard is partly supported by the Bézout Labex (New Monge Problems), funded by ANR, reference ANR-10-LABX-58.

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

# Appendix

## Table of Contents

# A  Additonnal Experiments

## A.1  Mini-batch DRAG.

As for Vanilla SGD, we can take advantage of GPU parallelization and replace the gradient estimator using one sample $X \sim \mu$

$$\nabla_{\mathbf{g}} h_\varepsilon(X, \mathbf{g})$$

by a mini-batch estimator, using $n_b \geq 1$ i.i.d samples $X_1, ..., X_{n_b}$ samples of the source measure at once

$$\frac{1}{n_b} \sum_{k=0}^{n_b} \nabla_{\mathbf{g}} h_\varepsilon(X_k, \mathbf{g}). \tag{7}$$

Of course, no matter the choice $n_b$, (7) defines an unbiased estimator of $\nabla H_\varepsilon(\mathbf{g})$.

Using a mini-batch of size $n_b$, we suggest multiplying $\gamma_1$ by $\sqrt{n_b}$, as is usual with mini-batch SGD. The following figure shows the acceleration due to mini-batching on Example 1, 2 and 3, while maintaining the same computational time when using a GPU. Indeed, each mini-batch estimator has an error an order of magnitude lower than the non-batched ones, even with a small mini-batch size of $n_b = 16$.

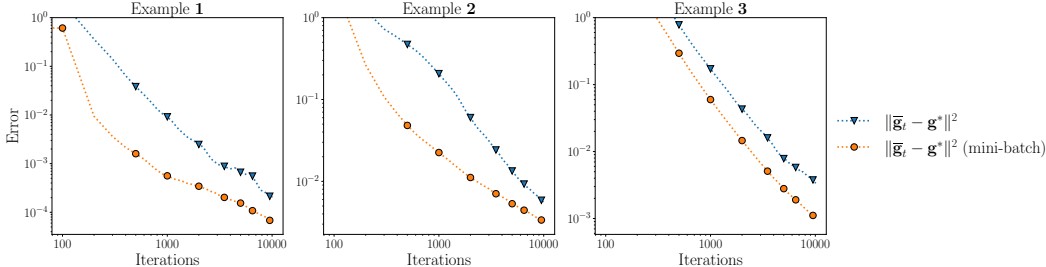

Figure 6: Comparison of the non mini-batched and mini-batched estimators on Example 1, 2 and 3, $n_b = 16$.

## A.2  Weighted Averaging: Maintaining a better trade-off between averaged and non-averaged iterations.

I is well known that the averaged algorithm can suffer from bad initialization. One strategy to overcome this is weighted averaging [39]. Namely, we replace the averaged estimator $\overline{\mathbf{g}}_t = \frac{1}{t+1} \sum_{k=0}^{t} \mathbf{g}_t$, by

$$\overline{\mathbf{g}}_t^{(\omega)} := \frac{1}{\sum_{k=0}^{t} \log(k+1)^\omega} \sum_{k=0}^{t} \log(k+1)^\omega \mathbf{g}_k,$$

with a parameter $\omega > 0$. The parameter $\omega$ balances the weights assigned to the estimators $\mathbf{g}_k$. As $\omega$ increases, greater importance is given to the more recent estimates, while we retrieve $\overline{\mathbf{g}}_t$ when $\omega$ goes to $0$. As for the usual averaged estimators, we can perform the weighted average online, without having to store all the iterates, with the recursion

$$\overline{\mathbf{g}}_{t+1}^{(\omega)} = \left(1 - \frac{\ln(t+1)^\omega}{\sum_{k=0}^{t} \ln(k+1)^\omega}\right) \overline{\mathbf{g}}_t^{(\omega)} + \frac{\ln(t+1)^\omega}{\sum_{k=0}^{t} \ln(k+1)^\omega} \mathbf{g}_{t+1}.$$

It is important to note that $\overline{\mathbf{g}}_t^{(\omega)}$ will have the same asymptotic convergence guarantees as $\overline{\mathbf{g}}_t$.

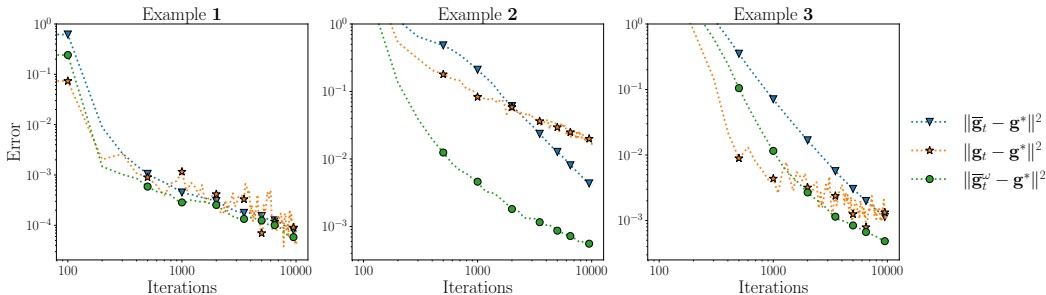

Figure 7: Comparison between $\mathbf{g}_t, \overline{\mathbf{g}}_t$ and $\overline{\mathbf{g}}_t^{(\omega)}$ on Examples 1, 2 and 3, with $\omega = 2$

As illustrated in Figure 7, the weighted average estimator consistently outperforms $\overline{\mathbf{g}}_t$, achieving orders of magnitude better performance in Examples 2 and 3. Note that a mini-batch size of 16 was used for all experiments, each repeated 10 times.

### A.3 DRAG compared to Adam.

We compare here the performance of our algorithm DRAG to that of the Adam algorithm [30], on Example 1, with $M \in 200, 2000$. The experiment was repeated 10 times. For this comparison, we fixed the parameters of DRAG to $(\sqrt{M}, 1/3, 2/3)$ and ran the algorithm for $t = 10^5$ iterations. The parameters for Adam were set to $\beta_1 = 0.9$, $\beta_2 = 0.999$, and $\lambda = 10^{-3}$ (learning rate/weight decay). As shown in Figure 8, DRAG clearly outperforms Adam on this example, particularly in the early iterations and as the number of points increases.

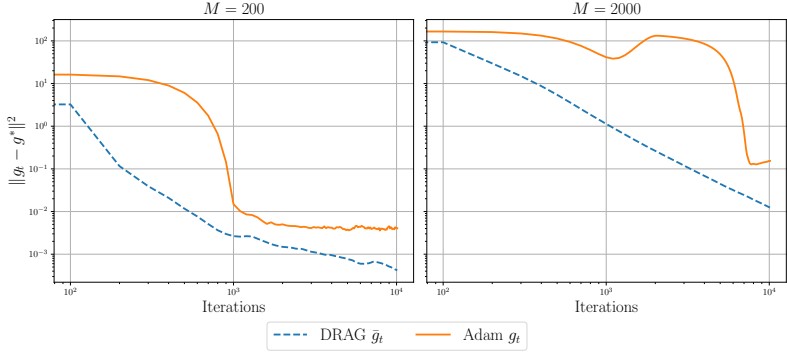

Figure 8: Comparison of DRAG with Adam on Example 1, for different values of $M$.

### A.4 DRAG compared to non regularized ASGD

As discussed in the numerical section, we observed empirically that all methods, including the non-regularized projected ASGD, eventually converge to the true solution given a sufficient number of iterations. In Figure 9, we report additional experiments with a larger iteration budget to confirm this behavior. Notably, even the non-regularized ASGD converges, albeit much more slowly. In contrast, DRAG converges significantly faster and achieves higher accuracy earlier in the optimization process. Among the DRAG variants, we observe that choices of $a \in 0.2, 0.4$ yield the best performance, which aligns with our theoretical analysis suggesting that $a = \frac{1}{3}^-$ achieves the optimal convergence rate in this setting, since $b = \frac{2}{3}$. These results reinforce our claim from the main text that, while all schemes converge given enough iterations, DRAG remains consistently one to two orders of magnitude more accurate due to its improved early-stage performance (see Appendix A).

Note also that the convergence of the non-regularized ASGD is encouraging, as it highlights that, with a sufficiently large number of steps, the effect of vanishing regularization is not a deterrent. Indeed, as $t \to \infty$, the regularized gradient becomes numerically indistinguishable from the non-regularized one, essentially corresponding to the difference between a tempered softmax with temperature $\varepsilon$ and

an argmax. This further supports the view that DRAG serves as an effective acceleration mechanism in the early stages of optimization and that by decreasing the regularization, we will not hit a plateau.

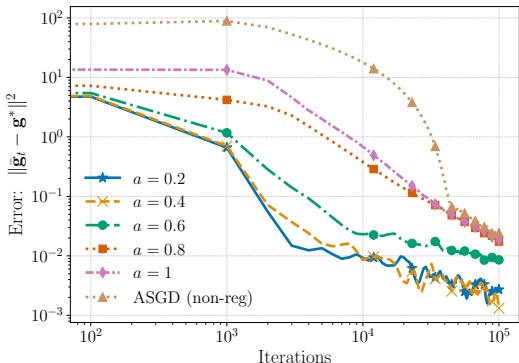

Figure 9: Comparison of DRAG with different $a$ and non-regularized ASGD

# B  Proofs

**Additionnal notations.**

For any $c > 0$ we define the function $t \mapsto \Psi_c(t)$ such that

$$\sum_{t=1}^{T} t^{-c} \leq \Psi_c(T) := \begin{cases} 1 + \ln(T+1) & \text{if } c = 1, \\ \frac{2c-1}{c-1} & \text{if } c > 1, \\ 1 + \frac{1}{1-c}(T+1)^{1-c} & \text{if } c < 1. \end{cases} \tag{8}$$

For a sequence $(u_t)_{t \in \mathbb{N}}$, if $\frac{t}{2} \notin \mathbb{N}$, $u_{\frac{t}{2}}$ must be understood as $u_{\lceil \frac{t}{2} \rceil}$.

In all the sequel, we note

$$\Delta_t = \|\mathbf{g}_t - \mathbf{g}_{\varepsilon_t}^*\|^2.$$

Remark that the dependence in $t$ is both in the estimator $\mathbf{g}_t$ and the optimizer $\mathbf{g}_{\varepsilon_t}^*$. We also recall that we note $D_{\mathcal{C}} := \sup_{\mathbf{g}, \mathbf{g}' \in \mathcal{C}} \|\mathbf{g} - \mathbf{g}'\| < \infty$ .

## B.1  Proof of Theorem 1: Convergence rate of the non averaged iterates.

*Proof.* Using Lemma 3, for any $t \geq t_{a,\alpha}$, we have

$$\Delta_{t+1} \leq \Delta_t - 2\gamma_{t+1} \left\langle \nabla_{\mathbf{g}} h_{\varepsilon_t}(\mathbf{g}_t, X_{t+1}), \mathbf{g}_t - \mathbf{g}_{\varepsilon_t}^* \right\rangle + 5\gamma_{t+1}^2.$$

Let $\mathcal{F}_t$ denote the filtration generated by the samples $X_1, \dots, X_t \overset{\text{iid}}{\sim} \mu$, that is $\mathcal{F}_t = \sigma(X_1, \dots, X_t)$ and taking the conditional expectation, we have

$$\mathbb{E}[\Delta_{t+1}|\mathcal{F}_t] \leq \Delta_t - 2\gamma_{t+1} \left\langle \nabla H_{\varepsilon_t}(\mathbf{g}_t), \mathbf{g}_t - \mathbf{g}_{\varepsilon_t}^* \right\rangle + 5\gamma_t^2. \tag{9}$$

Using Lemma 2 on the restricted strong convexity of $H_{\varepsilon_t}$, we have

$$\left\langle \nabla H_{\varepsilon_t}(\mathbf{g}_t), \mathbf{g}_t - \mathbf{g}_{\varepsilon_t}^* \right\rangle \geq \rho_*(1 - e^{-1}) \|\mathbf{g}_t - \mathbf{g}_{\varepsilon_t}^*\|^2 \mathbb{1}_{\|\mathbf{g}_t - \mathbf{g}_{\varepsilon_t}^*\| \leq \varepsilon_t/2} .$$

Therefore, we have

$$\mathbb{E}[\Delta_{t+1} \mid \mathcal{F}_t] \leq \left[1 - 2\rho_*(1 - e^{-1})\gamma_{t+1}\right] \Delta_t + \left[2\rho_*(1 - e^{-1})\mathbb{1}_{\|\mathbf{g}_t - \mathbf{g}_{\varepsilon_t}^*\| \geq \varepsilon_t/2}\right] \gamma_{t+1}\Delta_t + 5\gamma_{t+1}^2. \tag{10}$$

Using Proposition 2, for all $p$ and $\beta \in (0,1)$, there exists $C_{\beta,p}$ such that for all $t \geq 0$, $\mathbb{E}[\Delta_t^p] \leq C_{\beta,p} \frac{\gamma_t^p \varepsilon_t^{\beta p}}{\varepsilon_t^p} \leq C_{\beta,p} t^{-bp + a(1-\beta)p}$. Therefore,

$$
\begin{aligned}
\mathbb{E}[\mathbb{1}_{\|\mathbf{g}_t - \mathbf{g}_{\varepsilon_t}^*\| \geq \varepsilon_t/2}] &= \mathbb{E}[\mathbb{1}_{\|\mathbf{g}_t - \mathbf{g}_{\varepsilon_t}^*\|^{2p} \geq \varepsilon_t^{2p}/2^p}] \\
&\leq C_{\beta,p} t^{-bp + ap(3-\beta)} \qquad \text{by Markov's inequality} \\
&\leq C_{\beta, \frac{4b}{b-a(3-\beta)}} t^{-4b} \qquad \text{taking } p = \frac{4b}{b-a(3-\beta)}, \text{ with } a(3-\beta) < b. \quad (11)
\end{aligned}
$$

Note that, since $2a < b$, we can always choose $\beta$ such that the inequality $a(3-\beta) < b$ holds. Using the fact that $\Delta_t \leq D_{\mathcal{C}}^2$ and taking the expectation in (10), we obtain

$$
\mathbb{E}\left[\Delta_{t+1}\right] \leq \left[1 - 2\rho_*(1-e^{-1})\gamma_{t+1}\right] \mathbb{E}[\Delta_t] + 5\gamma_{t+1}^2 + C_{\beta, \frac{4b}{b-a(3-\beta)}} t^{-5b} D_{\mathcal{C}}^2.
$$

Let $t_\gamma := \min\{t, 2\lambda\gamma_{t+1} \leq 1\}$ and $t_0 := \max\{t_{a,\alpha}, t_\gamma\}$, we apply Proposition 5 to obtain

$$
\mathbb{E}\left[\Delta_t\right] \leq \exp\left(-2\lambda \sum_{i=t_0+1}^{t} \gamma_i\right)\left(D_{\mathcal{C}}^2 + \sum_{k=t_0}^{t} 5\gamma_k^2\right) + \frac{5}{2\lambda}\gamma_{\frac{t}{2}-1} + o(\gamma_t). \quad (12)
$$

Applying Corollary 2, the exponential product converges exponentially fast to 0 and an asymptotic comparison gives

$$
\mathbb{E}[\Delta_t] \leq \frac{5}{2\rho^*(1-e^{-1})}\gamma_{\frac{t}{2}-1} + o(\gamma_t) \lesssim \frac{\gamma_t}{\rho^*}.
$$

We conclude by using Proposition 1, using the bound $\|\mathbf{g}_{\varepsilon_t}^* - \mathbf{g}^*\| \lesssim \varepsilon_t^{1+\alpha'}$.

$\qquad\qquad\qquad\qquad\qquad\qquad\qquad\qquad\qquad\qquad\qquad\qquad\qquad\qquad\qquad\qquad\qquad\qquad\quad$ □

**Proposition 3.** *Under the same assumptions as in Theorem 1 , we have for any $\alpha' \in (0, \alpha)$*

$$
\mathbb{E}\left[\|\mathbf{g}_t - \mathbf{g}^*\|^2\right] \lesssim \frac{1}{\rho_*^2 \cdot t^{2b}} + \frac{1}{t^{4a+4a\alpha'}}, \qquad t \geq 1.
$$

**Remark:** Note that this proposition directly proves Theorem 1, but we decided to split them, to have a cleaner proof of Theorem 1.

*Proof.* We begin by squaring equation (13) of Lemma 3. For $t \geq t_{a,\alpha}$, where $t_{a,\alpha}$ is defined in (15), we have

$$
\begin{aligned}
\Delta_{t+1}^2 &\leq \left(\Delta_t - 2\gamma_{t+1}\left\langle \nabla_{\mathbf{g}} h_{\varepsilon_t}(\mathbf{g}_t, X_{t+1}), \mathbf{g}_t - \mathbf{g}_{\varepsilon_t}^*\right\rangle + 5\gamma_{t+1}^2\right)^2 \\
&\leq \Delta_t^2 + 4\gamma_{t+1}^2\left\langle \nabla_{\mathbf{g}} h_{\varepsilon_t}(\mathbf{g}_t, X_{t+1}), \mathbf{g}_t - \mathbf{g}_{\varepsilon_t}^*\right\rangle^2 + 25\gamma_{t+1}^4 \\
&\quad \underbrace{- 4\Delta_t\gamma_{t+1}\left\langle \nabla_{\mathbf{g}} h_{\varepsilon_t}(\mathbf{g}_t, X_{t+1}), \mathbf{g}_t - \mathbf{g}_{\varepsilon_t}^*\right\rangle}_{=:A} + 10\Delta_t\gamma_{t+1}^2 \underbrace{- 20\gamma_{t+1}^3\left\langle \nabla_{\mathbf{g}} h_{\varepsilon_t}(\mathbf{g}_t, X_{t+1}), \mathbf{g}_t - \mathbf{g}_{\varepsilon_t}^*\right\rangle}_{=:B}.
\end{aligned}
$$

Taking the conditional and using Lemma 2, we have

$$
\mathbb{E}[A \mid \mathcal{F}_t] \geq 4\Delta_t^2 \rho_*(1-e^{-1})\gamma_{t+1}\mathbb{1}_{\|\mathbf{g}_t - \mathbf{g}_{\varepsilon_t}\| \leq \varepsilon_t/2}.
$$

We also use the simple bound

$$
\mathbb{E}[B \mid \mathcal{F}_t] \geq 0.
$$

These two inequalities lead to

$$
\begin{aligned}
\mathbb{E}[\Delta_t^2 \mid \mathcal{F}_t] &\leq \left[1 - 4\rho_*(1-e^{-1})\gamma_{t+1}\right]\Delta_t^2 + 4\rho_*(1-e^{-1})\gamma_{t+1}\mathbb{1}_{\|\mathbf{g}_t - \mathbf{g}_{\varepsilon_t}\| \leq \varepsilon_t/2} \\
&\quad + 4\gamma_{t+1}^2\mathbb{E}\left[\left\langle \nabla h_{\varepsilon_t}(\mathbf{g}_t, X_{t+1}), \mathbf{g}_t - \mathbf{g}_{\varepsilon_t}^*\right\rangle^2 \mid \mathcal{F}_t\right] + 25\gamma_{t+1}^4 + 5\Delta_t\gamma_{t+1}^2.
\end{aligned}
$$

Using that the gradient norm is bounded by two, we apply the Cauchy-Schwarz inequality to obtain

$$4\gamma_{t+1}^2 \left\langle \nabla_{\mathbf{g}} h_{\varepsilon_t}(\mathbf{g}_t, X_{t+1}), \mathbf{g}_t - \mathbf{g}_{\varepsilon_t}^* \right\rangle^2 \leq 16\gamma_{t+1}^2 \|\mathbf{g}_t - \mathbf{g}_{\varepsilon_t}^*\|^2 \leq 16\Delta_t \gamma_{t+1}^2.$$

Applying Hölder's inequality yields

$$21\Delta_t\gamma_{t+1}^2 \leq \left( \Delta_t \sqrt{2\rho_*(1 - e^{-1})} \frac{1}{\sqrt{2\rho_*(1 - e^{-1})}} 21\gamma_{t+1} \right) \gamma_{t+1}$$

$$\leq \gamma_{t+1}\Delta_t^2 \rho_*(1 - e^{-1}) + \frac{21^2}{4\rho_*(1 - e^{-1})}\gamma_{t+1}^3.$$

Summing up these inequalities, we obtain

$$\mathbb{E}[\Delta_{t+1}^2 \mid \mathcal{F}_t] \leq \left[1 - 3\rho_*(1 - e^{-1})\gamma_{t+1}\right]\Delta_t^2 + 4\rho_*(1 - e^{-1})\gamma_{t+1}\mathbb{1}_{\|\mathbf{g}_t - \mathbf{g}_{\varepsilon_t}\| \geq \varepsilon_t/2}$$

$$+ \frac{21^2}{4\rho_*(1 - e^{-1})}\gamma_{t+1}^3 + 25\gamma_{t+1}^4.$$

Similarly to the case $p = 1$, using that $\mathbb{P}[\|\mathbf{g}_t - \mathbf{g}_{\varepsilon_t}^*\| \geq \varepsilon_t/2] \leq C_{\beta, \frac{4b}{b-a(3-\beta)}} t^{-4b}$ by (11) and that $\Delta_t^2 \leq D_{\mathcal{C}}^4$ for all $t$, taking the expectation yields

$$\mathbb{E}[\Delta_{t+1}^2] \leq (1 - 3\lambda\gamma_{t+1})\mathbb{E}[\Delta_t^2] + \frac{21^2}{4\lambda}\gamma_{t+1}^3 + 25\gamma_{t+1}^4 + 4\lambda C_{\beta, \frac{4b}{b-a(3-\beta)}} t^{-5b} D_{\mathcal{C}}^4.$$

Again, as in the case $p = 1$, applying Proposition 5 and Corollary 2 and using that $\|\mathbf{g}_{\varepsilon_t}^* - \mathbf{g}^*\| \lesssim \varepsilon_t^{1+\alpha'}$ concludes the proof. $\qquad\square$

**Lemma 3.** *Under the assumptions of Theorem 1, there exists a finite time $t_{a,\alpha}$, depending on $a$ and $\alpha$, such that for all $t \geq t_{a,\alpha}$, we have*

$$\Delta_{t+1} \leq \Delta_t - 2\gamma_{t+1}\left\langle \nabla_{\mathbf{g}} h_{\varepsilon_t}(\mathbf{g}_t, X_{t+1}), \mathbf{g}_t - \mathbf{g}_{\varepsilon_t}^* \right\rangle + 5\gamma_{t+1}^2. \tag{13}$$

*Proof.* By definition of the gradient step at time $t + 1$ and since $\mathbf{g}_{\varepsilon_{t+1}}^* \in \mathcal{C}$, we have

$$\Delta_{t+1} = \|\mathbf{g}_{t+1} - \mathbf{g}_{\varepsilon_{t+1}}^*\|^2$$

$$= \|\mathrm{Proj}_{\mathcal{C}}(\mathbf{g}_t - \gamma_{t+1}\nabla_{\mathbf{g}} h_{\varepsilon_t}(\mathbf{g}_t, X_{t+1})) - \mathbf{g}_{\varepsilon_{t+1}}^*\|^2$$

$$\leq \|\mathbf{g}_t - \gamma_{t+1}\nabla_{\mathbf{g}} h_{\varepsilon_t}(\mathbf{g}_t, X_{t+1}) - \mathbf{g}_{\varepsilon_{t+1}}^*\|^2.$$

Then, incorporating the change of optimum between time $t$ and $t + 1$, we get

$$\Delta_{t+1} \leq \|\mathbf{g}_t - \gamma_{t+1}\nabla_{\mathbf{g}} h_{\varepsilon_t}(\mathbf{g}_t, X_{t+1}) - \mathbf{g}_{\varepsilon_t}^* + \mathbf{g}_{\varepsilon_t}^* - \mathbf{g}_{\varepsilon_{t+1}}^*\|^2$$

$$\leq \|\mathbf{g}_t - \gamma_{t+1}\nabla_{\mathbf{g}} h_{\varepsilon_t}(\mathbf{g}_t, X_{t+1}) - \mathbf{g}_{\varepsilon_t}^*\|^2 + \|\mathbf{g}_{\varepsilon_t}^* - \mathbf{g}_{\varepsilon_{t+1}}^*\|^2$$

$$+ 2\left\langle \mathbf{g}_t - \gamma_{t+1}\nabla_{\mathbf{g}} h_{\varepsilon_t}(\mathbf{g}_t, X_{t+1}) - \mathbf{g}_{\varepsilon_t}^*, \mathbf{g}_{\varepsilon_t}^* - \mathbf{g}_{\varepsilon_{t+1}}^* \right\rangle.$$

Using Corollary 2.2 in [18] (see Proposition 1), there exists $K_0 > 0$ such that for any $\alpha' \in ]0, \alpha[$

$$\|\mathbf{g}_{\varepsilon_t}^* - \mathbf{g}_{\varepsilon_{t+1}}^*\| \leq K_0\varepsilon_t^{\alpha'}(\varepsilon_t - \varepsilon_{t+1}) \leq K_0 t^{-a\alpha'}\left(t^{-a} - (t+1)^{-a}\right) \leq aK_0 t^{-(1+a+a\alpha')}. \tag{14}$$

For clarity, we define $r_t := aK_0 t^{-(1+a+a\alpha')}$ and $R_t := (2D_{\mathcal{C}} + 2\gamma_{t+1} + r_t)r_t$.

Using that for all $t$, $\mathbf{g}_t \in \mathcal{C}$, and that for all $x \in \mathbb{R}^d, \mathbf{g} \in \mathbb{R}^M, \|\nabla_{\mathbf{g}} h_{\varepsilon_t}(\mathbf{g}, x)\| \leq 2$, we obtain

$$\Delta_{t+1} \leq \|\mathbf{g}_t - \gamma_{t+1}\nabla_{\mathbf{g}} h_{\varepsilon_t}(\mathbf{g}_t, X_{t+1}) - \mathbf{g}_{\varepsilon_t}^*\|^2 + (2D_{\mathcal{C}} + 2\gamma_{t+1})\|\mathbf{g}_{\varepsilon_t}^* - \mathbf{g}_{\varepsilon_{t+1}}^*\| + \|\mathbf{g}_{\varepsilon_t}^* - \mathbf{g}_{\varepsilon_{t+1}}^*\|^2$$

$$\leq \|\mathbf{g}_t - \gamma_{t+1}\nabla_{\mathbf{g}} h_{\varepsilon_t}(\mathbf{g}_t, X_{t+1}) - \mathbf{g}_{\varepsilon_t}^*\|^2 + R_t$$

$$\leq \|\mathbf{g}_t - \mathbf{g}_{\varepsilon_t}^*\|^2 - 2\gamma_{t+1}\left\langle \nabla_{\mathbf{g}} h_{\varepsilon_t}(\mathbf{g}_t, X_{t+1}), \mathbf{g}_t - \mathbf{g}_{\varepsilon_t}^* \right\rangle + \gamma_{t+1}^2\|\nabla_{\mathbf{g}} h_{\varepsilon_t}(\mathbf{g}_t, X_{t+1})\|^2 + R_t$$

$$\leq \Delta_t - 2\gamma_{t+1}\left\langle \nabla_{\mathbf{g}} h_{\varepsilon_t}(\mathbf{g}_t, X_{t+1}), \mathbf{g}_t - \mathbf{g}_{\varepsilon_t}^* \right\rangle + 4\gamma_{t+1}^2 + R_t.$$

Note that, since we have $1 + a + a\alpha > 2b$, we can also take $\alpha' \in ]0, \alpha[$ such that $1 + a + a\alpha' > 2b$. Consequently, the sequence $R_t/\gamma_t^2$ is decreasing and tends to 0. For conciseness, we note

$$t_{a,\alpha} := \min \left\{ t \geq 1 : R_t \leq \gamma_t^2 \right\}. \tag{15}$$

For any $t \geq t_{a,\alpha}$, we then obtain the following upper bound of $\Delta_{t+1}$ in terms of $\Delta_t$ and the gradient direction:

$$\Delta_{t+1} \leq \Delta_t - 2\gamma_{t+1} \left\langle \nabla_{\mathbf{g}} h_{\varepsilon_t}(\mathbf{g}_t, X_{t+1}), \mathbf{g}_t - \mathbf{g}_{\varepsilon_t}^* \right\rangle + 5\gamma_{t+1}^2.$$

$\square$

## B.2   Proof of Theorem 2: Convergence rate of DRAG

*Proof.* We start with a decomposition of the gradient step, similar to [26]. By abuse of notation, we note

$$\nabla_*^2 := \nabla^2 H_0(\mathbf{g}_{\varepsilon_k}^*)$$

and define the following differences:

$$
\begin{aligned}
p_k &:= \mathrm{Proj}_{\mathcal{C}} \left(\mathbf{g}_k - \gamma_{k+1}\nabla_{\mathbf{g}} h_{\varepsilon_k}(\mathbf{g}_k, X_{k+1})\right) - \left(\mathbf{g}_k - \gamma_{k+1}\nabla_{\mathbf{g}} h_{\varepsilon_k}(\mathbf{g}_k, X_{k+1})\right), \\
\xi_{k+1} &:= \nabla H_{\varepsilon_k}(\mathbf{g}_k) - \nabla_{\mathbf{g}} h_{\varepsilon_k}(\mathbf{g}_k, X_{k+1}), \\
\sigma_k &:= \nabla H_0(\mathbf{g}_k) - \nabla H_{\varepsilon_k}(\mathbf{g}_k) \\
\delta_k &:= \nabla H_0(\mathbf{g}_k) - \nabla_*^2 \left(\mathbf{g}_k - \mathbf{g}_0^*\right).
\end{aligned}
$$

The term $p_k$ represents the difference between the projected and non-projected steps. Note that $p_k = 0$ if $\mathbf{g}_k - \gamma_{k+1}\nabla_{\mathbf{g}} h_{\varepsilon_k}(\mathbf{g}_k, X_{k+1}) \in \mathcal{C}$. The term $\xi_k$ is a martingale difference $\xi_k$ representing the difference between the regularized gradient and its non-biased estimator. $\sigma_k$ represents the difference between the $\varepsilon_k$-regularized gradient and the non-regularized gradient. Finally, $\delta_k$ represents the difference between the gradient at $\mathbf{g}_k$ and its linear approximation given by the Hessian at the optimum.

Let $I_M$ denote identity matrix of $\mathcal{M}_M(\mathbb{R})$, observe that for any $k \in \mathbb{N}$

$$
\begin{aligned}
\mathbf{g}_{k+1} - \mathbf{g}_0^* &= \mathrm{Proj}_{\mathcal{C}} \left(\mathbf{g}_k - \gamma_{k+1}\nabla_{\mathbf{g}} h_{\varepsilon_k}(\mathbf{g}_k, X_{k+1})\right) - \mathbf{g}_0^* \\
&= \mathbf{g}_k - \gamma_{k+1}\nabla_{\mathbf{g}} h_{\varepsilon_k}(\mathbf{g}_k, X_{k+1}) - \mathbf{g}_0^* - p_k && \text{incorporating } p_k \\
&= \mathbf{g}_k - \gamma_{k+1}\nabla H_{\varepsilon_k}(\mathbf{g}_k) - \mathbf{g}_0^* + \gamma_{k+1}\xi_{k+1} - p_k && \text{incorporating } \xi_{k+1} \\
&= \mathbf{g}_k - \gamma_{k+1}\nabla H_0(\mathbf{g}_k) + \gamma_{k+1}\sigma_k - \mathbf{g}_0^* + \gamma_{k+1}\xi_{k+1} - p_k && \text{incorporating } \sigma_k \\
&= \left(I_M - \gamma_{k+1}\nabla_k^2\right)(\mathbf{g}_k - \mathbf{g}_0^*) - \gamma_{k+1}\delta_k + \gamma_{k+1}\sigma_k + \gamma_{k+1}\xi_{k+1} + p_k. \\
& && \text{incorporating } \delta_k
\end{aligned}
$$

Thus, we have that

$$\nabla_*^2 (\mathbf{g}_k - \mathbf{g}_0^*) = \frac{\mathbf{g}_k - \mathbf{g}_{k+1}}{\gamma_{k+1}} - \delta_k + \sigma_k + \xi_{k+1} + \frac{p_k}{\gamma_{k+1}}.$$

Observe that there is an orthogonal matrix $U$ such that $\nabla_*^2 = U \mathrm{diag}(\lambda_1, \ldots, \lambda_{M-1}, 0) U^\top$. Therefore, in the following, we denote

$$\left(\nabla^2\right)^{-1} = U \mathrm{diag}\left(\lambda_1^{-1}, \ldots, \lambda_{M-1}^{-1}, 0\right) U^\top$$

the inverse of $\nabla_*^2$, restricted to the subspace $\mathrm{Vect}(\mathbf{1}_M)^\perp$. Note that we have [18, Theorem 3.2]

$$\min_{j \in [\![1, M-1]\!]} \lambda_j \geq \rho_*, \quad \text{for all } k \geq 0.$$

Taking all the equalities in $\mathrm{Vect}(\mathbf{1}_M)^\perp$, that is, considering all our vectors in the subspace $\mathrm{Vect}(\mathbf{1}_M)^\perp$, we have

$$
\begin{aligned}
(\overline{\mathbf{g}}_t - \mathbf{g}_0^*) &= \frac{1}{t+1}\sum_{k=0}^{t} \left(\nabla_*^2\right)^{-1} \frac{\mathbf{g}_k - \mathbf{g}_{k+1}}{\gamma_{k+1}} - \frac{1}{t+1}\sum_{k=0}^{t} \left(\nabla_*^2\right)^{-1} \delta_k \\
&\quad + \frac{1}{t+1}\sum_{k=0}^{t} \sigma_k + \frac{1}{t+1}\sum_{k=0}^{t} \left(\nabla_*^2\right)^{-1} \xi_{k+1} + \frac{1}{t+1}\sum_{k=0}^{t} \left(\nabla_*^2\right)^{-1} \frac{p_k}{\gamma_{k+1}}.
\end{aligned}
$$

We will now give the convergence rate for each sum. Note that thanks to the introduction of $\sigma_k$, we will directly be able to use the local smoothness and strong convexity of $H_0$, proved in our setting in [32].

- **Convergence rate for** $\frac{1}{t+1}\sum_{k=0}^{t}\frac{\mathbf{g}_k - \mathbf{g}_{k+1}}{\gamma_{k+1}}$.

$$
\begin{aligned}
\sum_{k=0}^{t}\frac{\mathbf{g}_k - \mathbf{g}_{k+1}}{\gamma_{k+1}} &= \sum_{k=0}^{t}\frac{(\mathbf{g}_k - \mathbf{g}^*) - (\mathbf{g}_{k+1} - \mathbf{g}^*)}{\gamma_{k+1}} \\
&= \sum_{k=0}^{t}\frac{\mathbf{g}_k - \mathbf{g}^*}{\gamma_{k+1}} - \sum_{k=0}^{t}\frac{\mathbf{g}_{k+1} - \mathbf{g}^*}{\gamma_{k+1}} \\
&= \sum_{k=1}^{t}\left(\frac{1}{\gamma_{k+1}} - \frac{1}{\gamma_k}\right)(\mathbf{g}_k - \mathbf{g}^*) + \frac{\mathbf{g}_0 - \mathbf{g}^*}{\gamma_1} - \frac{\mathbf{g}_{t+1} - \mathbf{g}^*}{\gamma_{t+1}}.
\end{aligned}
$$

Remark that $\gamma_{t+1}^{-1} - \gamma_t^{-1} \le 2\gamma_1^{-1}n^{b-1}$. By Theorem 1 (non-averaged iterates), $\mathbb{E}\left[\|\mathbf{g}_n - \mathbf{g}^*\|_v^2\right] \lesssim \frac{\gamma_1}{\rho_*}(t+1)^{-b}$. Therefore

$$
\mathbb{E}\left[\left\|\sum_{k=0}^{t}\frac{\mathbf{g}_k - \mathbf{g}_{k+1}}{\gamma_{k+1}}\right\|_v^2\right]^{\frac{1}{2}} \lesssim \frac{1}{\rho_*}\Psi_{1-b/2}(t+1) + D_{\mathcal{C}}\gamma_1^{-1} + \frac{1}{\sqrt{\gamma_1}\rho_*}(t+1)^{b/2}.
$$

We thus have the convergence rate

$$
\frac{1}{t+1}\mathbb{E}\left[\left\|\sum_{k=0}^{t}\frac{\mathbf{g}_k - \mathbf{g}_{k+1}}{\gamma_{k+1}}\right\|_v^2\right]^{\frac{1}{2}} \lesssim \frac{1}{\rho_*(t+1)^{1-b/2}}.
$$

- **Convergence rate for** $\frac{1}{t+1}\sum_{k=0}^{t}\delta_k$.

By [32, Theorem 1.3], there exists a ball $B(\mathbf{g}^*, d_1)$ with $d_1 > 0$ where $H$ is $\alpha$-Hölder. Therefore, by applying a Taylor expansion of $\nabla H_0(\mathbf{g})$ around $\mathbf{g}^*$, if $\mathbf{g}_k \in B(\mathbf{g}^*, d_1)$, we have

$$
\|\delta_k\| \lesssim \|\mathbf{g}_k - \mathbf{g}^*\|_v^{1+\alpha}.
$$

Otherwise, since the Hessian $H_0$ is uniformly bounded [32, Theorem 1.1], there exists a constant $C_\delta$ such that for any $\mathbf{g} \in \mathcal{C}$, $\|\nabla H(\mathbf{g}) - \nabla^2 H(\mathbf{g}^*)(\mathbf{g} - \mathbf{g}^*)\| \le C_\delta$.

Since $\mathbb{P}(\mathbf{g}_k \notin B(\mathbf{g}^*, d_1)) = \mathbb{P}(\|\mathbf{g}_k - \mathbf{g}_*\| > d_1)$, we obtain by Markov's inequality

$$
\begin{aligned}
\mathbb{E}[\|\delta_k\|_v^2] &= \mathbb{E}[\|\delta_k\|_v^2\mathbf{1}_{\mathbf{g}_k \in B(\mathbf{g}^*, d_1)}] + \mathbb{E}[\|\delta_k\|_v^2\mathbf{1}_{\mathbf{g}_k \notin B(\mathbf{g}^*, d_1)}] \\
&\lesssim \mathbb{E}\left[\|\mathbf{g}_k - \mathbf{g}^*\|_v^{2+2\alpha}\right] + \frac{C_\delta^2}{d_1^{2+2\alpha}}\mathbb{E}[\|\mathbf{g}_n - \mathbf{g}^*\|_v^{2+2\alpha}] \\
&\lesssim \mathbb{E}[\|\mathbf{g}_k - \mathbf{g}^*\|_v^{2+2\alpha}].
\end{aligned}
$$

Therefore, using Minkowski's inequality, we have

$$
\begin{aligned}
\frac{1}{t+1}\mathbb{E}\left[\left\|\sum_{k=0}^{t}\delta_k\right\|_v^2\right]^{\frac{1}{2}} &\lesssim \frac{1}{t+1}\sum_{k=0}^{t}\frac{1}{\rho_*^{\frac{1+\alpha}{2}}}\gamma_{k+1}^{\frac{1+\alpha}{2}} \\
&\le \frac{1}{\rho_*^{\frac{1+\alpha}{2}}(t+1)}\Psi_{\frac{b+\alpha b}{2}} \\
&\lesssim \frac{1}{\rho_*^{\frac{1+\alpha}{2}}(t+1)^{\frac{b+\alpha b}{2}}}.
\end{aligned}
$$

• **Convergence rate for** $\frac{1}{t+1}\sum_{k=0}^{t}\xi_{k+1}$.

We recall that $\xi_{k+1} = \nabla H(\mathbf{g}_k) - \nabla_{\mathbf{g}}h(\mathbf{g}_k, X_{k+1})$ and thus $\mathbb{E}[\xi_{k+1}] = 0$.

Observe that $\mathbb{E}\left[\left\langle \sum_{k=0}^{n-1}\xi_{k+1}, \xi_{t+1}\right\rangle_v\right] = \mathbb{E}\left[\left\langle \sum_{k=0}^{n-1}\xi_{k+1}, \mathbb{E}\left[\xi_{t+1}|\mathcal{F}_t\right]\right\rangle_v\right] = 0$.

Thus, since $\mathbb{E}\left[\|\xi_k\|^2\right] \leq 4$ for all $k$, we have the convergence rate

$$\frac{1}{t+1}\mathbb{E}\left[\left\|\sum_{k=0}^{t}\xi_{k+1}\right\|_v^2\right]^{\frac{1}{2}} \leq \frac{2}{\sqrt{t+1}}.$$

• **Convergence rate of** $\frac{1}{t+1}\sum_{k=0}^{t}\sigma_k$.

Using Proposition 4, we have uniformly in $\mathbf{g}_k \in \mathcal{C}$ that, for all $\alpha' \in (0, \alpha)$,

$$\|\sigma_k\| = \|\nabla H_0(\mathbf{g}_k) - \nabla H_{\varepsilon_k}(\mathbf{g}_k)\| \lesssim \varepsilon^{1+\alpha'} \lesssim t^{a+a\alpha'}.$$

Therefore

$$\frac{1}{t+1}\left\|\sum_{k=0}^{t}\sigma_k\right\| \lesssim \frac{1}{t+1}\Psi_{a+a\alpha'}(t)$$

$$\lesssim \frac{1}{t^{a+a\alpha'}}.$$

• **Convergence rate for** $\frac{1}{t+1}\sum_{k=0}^{t}\frac{p_k}{\gamma_k}$.

Take $d_0$ such that $B(\mathbf{g}^*, d_0) \subset \mathcal{C}$. Defining $\nabla_k := \nabla_{\mathbf{g}}h(\mathbf{g}_k, X_{k+1})$ for conciseness, we obtain

$$\mathbb{E}\left[\|p_k\|_v^2\right] = \mathbb{E}\left[\|\mathrm{Proj}_{\mathcal{C}}(\mathbf{g}_k - \gamma_{k+1}\nabla_k) - (\mathbf{g}_k - \gamma_{k+1}\nabla_k)\|_v^2\right]$$

$$= \mathbb{E}\left[\|\mathrm{Proj}_{\mathcal{C}}(\mathbf{g}_k - \gamma_{k+1}\nabla_k) - (\mathbf{g}_k - \gamma_{k+1}\nabla_k)\|_v^2 \mathbf{1}_{\mathbf{g}_k - \gamma_{k+1}\nabla_k \notin \mathcal{C}}\right]$$

Since for any $y \in \mathcal{C}$, one has $\|x - \mathrm{Proj}_{\mathcal{C}}(x)\|_v \leq \|x - y\|_v$, taking $y = \mathbf{g_k}$, and since $\mathbf{g}_k - \gamma_{k+1}\nabla_k \notin \mathcal{C}$ is satisfied only if $\|\mathbf{g}_k - \gamma_{k+1}\nabla_k - \mathbf{g}^*\|_v > d_0$, we have

$$\mathbb{E}\left[\|p_k\|_v^2\right] \leq \mathbb{E}\left[\|\gamma_{k+1}\nabla_k\|_v^2 \mathbf{1}_{\|\mathbf{g}_k - \gamma_{k+1}\nabla_k - \mathbf{g}^*\|_v > d_0}\right]$$

$$\leq 4\gamma_{k+1}^2 \frac{\mathbb{E}\left[\|\mathbf{g}_k - \gamma_{k+1}\nabla_k - \mathbf{g}^*\|_v^4\right]}{d_0^4}$$

$$\leq \frac{\gamma_{k+1}^2}{d_0^4}\left(2^5\mathbb{E}\left[\|\mathbf{g}_k - \mathbf{g}^*\|_v^4\right] + 2^9\gamma_{k+1}^4\right)$$

$$\lesssim \frac{1}{\rho_*^2}\gamma_{k+1}^4$$

We thus have

$$\frac{1}{t+1}\mathbb{E}\left[\left\|\sum_{k=0}^{t}\frac{p_k}{\gamma_k}\right\|_v^2\right]^{\frac{1}{2}} \lesssim \frac{1}{t+1}\sum_{k=0}^{t}\frac{\gamma_{k+1}}{\rho_*}$$

$$\lesssim \frac{\gamma_1}{\rho_*(t+1)^b}.$$

• **Conclusion.**

Finally, summing up all the convergence rates, using Cauchy-Schwarz inequality and that $(A+B)^2 \leq 2(A^2+B^2)$ for any $A, B \in \mathbb{R}$ we obtain

$$\mathbb{E}\left[\|\nabla^2 H(\mathbf{g}^*)(\bar{\mathbf{g}}_n - \mathbf{g}^*)\|_v^2\right] \lesssim \frac{1}{\rho_*^2(t+1)^{2-b}} + \frac{1}{t^{2a+2a\alpha'}} + \frac{1}{\rho_*^{\frac{1+\alpha}{\alpha}}(t+1)^{b+\alpha b}} + \frac{1}{t+1} + \frac{\gamma_1^4}{\rho_*^2(t+1)^{2b}}.$$

Since $b > 2a$ and $b + \alpha a > 2a + 2a\alpha'$ and so noting $s = \min\{1, 2a + 2a\alpha'\}$, and since the Hessian norm is uniformly bounded, we finally obtain

$$\mathbb{E}\left[\|\bar{\mathbf{g}}_n - \mathbf{g}^*\|_v^2\right] \lesssim \frac{1}{t^s}.$$

$\square$

## B.3 Proof of Theorem 3: Convergence of the OT map estimator

*Proof.* We show that the convergence rate of $\bar{\mathbf{g}}_t$ to $\mathbf{g}_0^*$ implies a convergence rate a convergence rate for the map estimation. The Brenier map is given by $T_{\mu,\nu}(x) = x - \nabla(\mathbf{g}_0^*)^c(x)$; see for instance [46], Theorem 1.17. We thus focus on the convergence of $\nabla \bar{\mathbf{g}}_t^c$ to $\nabla(\mathbf{g}_0^*)^c$.

For all $j \in [\![1, M]\!]$, if $x$ lies in the interior of $\mathbb{L}_j(\mathbf{g})$, we have

$$\nabla \mathbf{g}^c(x) = x - y_j. \tag{16}$$

Therefore, given $\mathbf{g}, \mathbf{g}' \in \mathbb{R}^M$, if there exists a $j \in [\![1, M]\!]$ such that $x$ is the interior of $\mathbb{L}_j(\mathbf{g}) \cap \mathbb{L}_j(\mathbf{g}')$ we have

$$\nabla \mathbf{g}^c(x) = \nabla(\mathbf{g}')^c(x).$$

We will now follow arguments from [46], Section 6.4.2. Fix $j, j' \in [\![1, M]\!]$ such that $j \neq j'$ and $x$ is in the interior of $\mathbb{L}_j(\mathbf{g}) \cap \mathbb{L}_{j'}(\mathbf{g}')$. By definition of the $c$-transform, we observe that $\mathbb{L}_j(\mathbf{g})$ is defined by $M - 1$ linear inequalities of the form

$$\langle x, y_{j'} - y_j \rangle \leq a_{\mathbf{g}}(j, j') := g_j - g_{j'} + \frac{1}{2}\|y_{j'}\|_2^2 - \frac{1}{2}\|y_j\|_2^2.$$

Similarly, interchanging the role of $\mathbf{g}, \mathbf{g}'$ and $j, j'$ we have

$$\langle x, y_j - y_{j'} \rangle \leq a_{\mathbf{g}'}(j', j) := g'_{j'} - g'_j + \frac{1}{2}\|y_j\|_2^2 - \frac{1}{2}\|y_{j'}\|_2^2.$$

We obtain that

$$\mathbb{L}_j(\mathbf{g}) \cap \mathbb{L}_{j'}(\mathbf{g}') \subset \{x \in \mathbb{R}^d : -a_{\mathbf{g}'}(j', j) \leq \langle x, y_{j'} - y_j \rangle \leq a_{\mathbf{g}}(j, j')\}.$$

Moreover, noting $h = (h_1, ..., h_M) = \mathbf{g} - \mathbf{g}'$, we see that

$$|a_{\mathbf{g}'}(j', j) + a_{\mathbf{g}}(j, j')| \leq |h_{j'} - h_j|. \tag{17}$$

We have

$$\mu\left(\mathcal{A} := \left\{x \in \mathbb{R}^d, \nabla\mathbf{g}^c(x) \neq \nabla(\mathbf{g}')^c(x)\right\}\right)$$

$$= \mu\left(\bigcup_{j<j'} \mathbb{L}_j(\mathbf{g}) \cap \mathbb{L}_{j'}(\mathbf{g}')\right)$$

$$\leq \sum_{j<j'} \mu\left(\mathbb{L}_j(\mathbf{g}) \cap \mathbb{L}_{j'}(\mathbf{g}')\right)$$

$$\leq \sum_{j<j'} \mu\left(\{x \in \mathbb{R}^d : -a_{\mathbf{g}'}(j', j) \leq \langle x, y_{j'} - y_j \rangle \leq a_{\mathbf{g}}(j, j')\}\right).$$

Under Assumption 1, $\mu$ is a measure such that $\text{Supp}(\mu) \subset B(0, R)$ and it admits a density $d\mu$ bounded by $d\mu_{\max}$. Thus,

$$\mu(\mathcal{A}) \leq d\mu_{\max} \sum_{j<j'} \lambda_{\mathbb{R}^d}(\{x \in B(0, R) : -a_{\mathbf{g}'}(j', j) \leq \langle x, y_{j'} - y_j \rangle \leq a_{\mathbf{g}}(j, j')\})$$

$$\leq d\mu_{\max} \sum_{j<j'} \lambda_{\mathbb{R}^d}\left(\left\{x \in B(0, R) : -\frac{a_{\mathbf{g}'}(j', j)}{\|y_{j'} - y_j\|_2} \leq \left\langle x, \frac{y_{j'} - y_j}{\|y_{j'} - y_j\|_2}\right\rangle \leq \frac{a_{\mathbf{g}}(j, j')}{\|y_{j'} - y_j\|_2}\right\}\right)$$

$$\leq d\mu_{\max} \sum_{j<j'} \lambda_{\mathbb{R}^d}\left(\left\{x \in B(0, R) : -\frac{a_{\mathbf{g}'}(j', j)}{\|y_{j'} - y_j\|_2} \leq x_1 \leq \frac{a_{\mathbf{g}}(j, j')}{\|y_{j'} - y_j\|_2}\right\}\right),$$

by the rotational invariance of the Lebesgue measure. Combining this remark with (17) yields

$$\mu(\mathcal{A}) \leq \mathrm{d}\mu_{\max} R^{d-1} \sum_{j<j'} \frac{|h_{j'} - h_j|}{\|y_{j'} - y_j\|_2} .$$

Similarly, for the $L^p$ norm of the map difference, we obtain

$$\left\| \left\| \left( \nabla \mathbf{g}^c(\cdot) - \nabla(\mathbf{g}')^c(\cdot) \right) \right\|_q \right\|_{L^p(\mu)}^p \leq \sum_{j<j'} \int_{\mathbb{L}_j(\mathbf{g}) \cap \mathbb{L}_{j'}(\mathbf{g}')} \left\| \left( \nabla \mathbf{g}^c(\cdot) - \nabla(\mathbf{g}')^c(\cdot) \right) \right\|_q d\mu(x)$$

$$\leq \sum_{j<j'} \|y_{j'} - y_j\|_q \mu \left( \mathbb{L}_j(\mathbf{g}) \cap \mathbb{L}_{j'}(\mathbf{g}') \right)$$

$$\leq \mathrm{d}\mu_{\max} R^{d-1} \sum_{j<j'} \frac{\|y_{j'} - y_j\|_q |h_{j'} - h_j|}{\|y_{j'} - y_j\|_2}$$

$$\leq \mathrm{d}\mu_{\max} M^{(2-q)_+/2q} R^{d-1} 2M \|h\|_1 .$$

So, in particular, there exists a constant $C_\Delta > 0$, independent of the location of the points $y_j$, which grows at least linearly in $M$ such that

$$\left\| \left\| \left( \nabla \mathbf{g}^c(\cdot) - \nabla(\mathbf{g}')^c(\cdot) \right) \right\|_q \right\|_{L^p(\mu)}^p \leq C_\Delta \|\mathbf{g} - \mathbf{g}'\|_1 \leq C_\Delta \sqrt{M} \|\mathbf{g} - \mathbf{g}'\|.$$

Plugging in the convergence rate of $\overline{\mathbf{g}}_t$ to $\mathbf{g}^*$ concludes the proof. □

### B.4  Proof of Corollary 1: OT cost estimation

*Proof.* For any vector $\mathbf{g} \in \mathbb{R}^M$, we recall the definition of $\mathbb{L}(\mathbf{g}) = \bigcup_{j=1}^M \mathbb{L}_j(\mathbf{g})$ :

$$\text{for all } j \in [\![1, M]\!], \; \mathbb{L}_j(\mathbf{g}) := \left\{ x \in \mathbb{R}^d; \mathbf{g}^c(x) = \frac{1}{2} \|x - y_j\|_2^2 - g_j \right\} .$$

Note that $\mathbb{L}(\mathbf{g})$ defines a partition of $\mathbb{R}^d$ up to $\mu$-null sets , i.e. $\mu \left( \mathbb{L}_i(\mathbf{g}) \cap \mathbb{L}_j(\mathbf{g}) \right) = 0$ when $i \neq j$, and the convex sets $\mathbb{L}_j(\mathbf{g})$ are called power or Laguerre cells. We define the set

$$\mathcal{K}^\delta := \left\{ \mathbf{g} : \mathbb{R}^M \to \mathbb{R} \mid \forall i \in [\![1, M]\!], \mu \left( \mathbb{L}_i(\mathbf{g}) \right) > \delta \right\} .$$

Using Theorem 4.1 in [32], under Assumption 1, $H_0$ is uniformly $C^{2,\alpha}$ on $\mathcal{K}^\delta$. That is, there exists a constant $L$ such that $H_0$ is $L$-smooth on $\mathcal{K}^\delta$. Note that the constant $L$ depends on $\mu_{\min}, \delta, R$. We refer to [32], Remark 4.1 for more details.

By the first order condition, as soon as $\delta \leq w_{\min}$, we have $\mathbf{g}^* \in \mathcal{K}^\delta$. Indeed, at the optimum, we have for all $i \in [\![1, M]\!], \mu \left( \mathbb{L}_i(\mathbf{g}^*) \right) = w_i$. We fix here $\delta = \frac{1}{10} w_{\min}$.

Thanks to the $L$-smoothness, for any $\mathbf{g} \in \mathcal{K}^\delta$, we have

$$|H_0(\mathbf{g}) - H_0(\mathbf{g}^*)| \leq \frac{L}{2} \|\mathbf{g} - \mathbf{g}^*\|^2.$$

Note that, for any $\mathbf{g} \in \mathbb{R}^M$ and $i \in [\![1, M]\!]$, the difference of measure of the Laguerre cells $\mathbb{L}_i(\mathbf{g})$ and $\mathbb{L}_i(\mathbf{g}^*)$ is at most linear with respect to $\|\mathbf{g} - \mathbf{g}^*\|_\infty$. We refer to Theorem 3 or Section 6.4.2 in [46] for more details.

Therefore, there exists a constant $C_L$ such that, as soon as $\|\mathbf{g} - \mathbf{g}^*\|^2 \leq C_L$, we have that $\mathbf{g} \in K^\delta$. This constant depends on $\delta, \mu_{\max}, R$ and $d$ as in Theorem 3. Using Theorem 2, $\mathbb{E}[\|\overline{\mathbf{g}}_t - \mathbf{g}^*\|^2] = \mathcal{O}(t^{-s})$ with $s > 0$. Then

$$\mathbb{E}\left[ |H_0(\overline{\mathbf{g}}_t) - H_0(\mathbf{g}^*)| \right] = \mathbb{E}\left[ |H_0(\overline{\mathbf{g}}_t) - H_0(\mathbf{g}^*)| \mathbb{1}_{\overline{\mathbf{g}}_t \in K^\delta} \right] + \mathbb{E}\left[ |H_0(\overline{\mathbf{g}}_t) - H_0(\mathbf{g}^*)| \mathbb{1}_{\overline{\mathbf{g}}_t \notin K^\delta} \right]$$

$$\leq \frac{L}{2} \mathbb{E}[\|\overline{\mathbf{g}}_t - \mathbf{g}^*\|^2] + \max_{\mathbf{g} \in \mathcal{C}} |H_0(\mathbf{g}) - H_0(\mathbf{g}^*)| \mathbb{E}[\mathbb{1}_{\overline{\mathbf{g}}_t \notin K^\delta}]$$

$$\leq \frac{L}{2} \mathbb{E}[\|\overline{\mathbf{g}}_t - \mathbf{g}^*\|^2] + \max_{\mathbf{g} \in \mathcal{C}} |H_0(\mathbf{g}) - H_0(\mathbf{g}^*)| \mathbb{E}[\mathbb{1}_{\|\overline{\mathbf{g}}_t - \mathbf{g}^*\|^2 > C_L}]$$

$$= \mathcal{O} \left( \mathbb{E}[\|\overline{\mathbf{g}}_t - \mathbf{g}^*\|^2] \right) ,$$

where the Markov inequality of order 1 was used on $\mathbb{E}[\mathbb{1}_{\|\overline{\mathbf{g}}_t - \mathbf{g}^*\|^2 > C_L}]$. □

## B.5 Proof of Proposition 2: High probability being in $B(\mathbf{g}^*_{\varepsilon_t}, \varepsilon_t)$.

*Proof.* The proof of this proposition relies heavily on the technical Lemma 4, which we state and prove immediately after this proof.

We start with a base case at $\delta = u_0 = 0$, which provides an initial convergence rate for $\mathbb{E}[\Delta_t^p]$. Then, by an inductive argument, we gradually increase $u_n$ to improve this rate till the limit when $n$ tends to infinity, namely $\min\{b - a, a\}$.

**Base case ($u_0 = 0$).** Using Lemma 4, with $\lambda_{c,t} = \rho_* \frac{1 - e^{-4C_\infty}}{4C_\infty} \varepsilon_t$ if $c = 0$, and $\lambda_{c,t} = \rho_*(1 - e^{-1})\varepsilon_t\, t^c$ if $c \in (0, a]$, we have

$$\mathbb{E}[\Delta_{t+1}^p] \leq \mathbb{E}[\Delta_t^p]\left(1 - \gamma_{t+1}\lambda_{0,t} + C_{1,p}\,\gamma_{t+1}^2\right) + C_{2,p}\,\lambda_{c,t}^{-p+1}\,\gamma_{t+1}^{p+1}.$$

By applying Proposition 5 and Corollary 2, we obtain the following baseline convergence rate, for all $p > 0$:

$$\mathbb{E}[\Delta_t^p] \lesssim \frac{\gamma_t^p}{\varepsilon_t^p}.$$

**Inductive step (improving the rate).** Suppose that for some $u_n \in [0, \min\{b - a, a\})$, we already have

$$\mathbb{E}[\Delta_t^p] \lesssim t^{-p\,(b-a+u_n)}.$$

Choose $c < \frac{b-a+u_n}{2}$ and set $d = b - a + u_n - 2c > 0$, which is positive by construction. By Markov's inequality, we then get for all $q > 0$

$$\mathbb{P}\left[\Delta_t \geq t^{-2c}\right] = \mathbb{P}\left[\Delta_t^q \geq t^{-2qc}\right] \lesssim t^{-q\,(b-a+u_n-2c)} = t^{-dq}.$$

We take $q$ chosen large enough so that $dq > p + 1$.

Consequently, applying Lemma 4,

$$\mathbb{E}[\Delta_t^p] \leq \mathbb{E}[\Delta_t^p]\left(1 - \gamma_{t+1}\,\lambda_{c,t} + C_{1,p}\,\gamma_{t+1}^2\right) + C_{2,p}\,\lambda_{c,t}^{-p+1}\,\gamma_{t+1}^{p+1} + o\big(\gamma_{t+1}^{p+1}\big).$$

Therefore, if we pick any $u_{n+1} \in \big(0, \frac{b-a+u_n}{2}\big)$, applying Proposition 5 and Corollary 2, we see that

$$\mathbb{E}[\Delta_t^p] \lesssim \frac{\gamma_t^p}{\varepsilon_t^p}\, t^{-cp} \lesssim t^{-pu_{n+1}}.$$

As soon as $b - a > u_n$, we have $(b - a + u_n)/2 > u_n$ as a valid range upper range for $u_{n+1}$, so we can take $u_{n+1} > u_n$ and strictly improve our convergence rate.

**Achievability for all $\delta \in [0, \min\{b - a, a\})$.** Finally, note that the sequence defined by $u_0 = 0$ and $u_{n+1} = \frac{b-a+u_n}{2}$ converges to $(b - a)$, showing that every value $\delta$ up to $(b - a)$ can be reached through successive improvements. Since $c = a$ is the upper bound in Lemma 4, we can continue the limit $\min\{b - a, a\}$, so for all $\delta \in [0, \min\{b - a, a\})$, we have

$$\mathbb{E}\left[\Delta_t^p\right] \lesssim \frac{\gamma_t^p\, t^\delta}{\varepsilon_t^p}.$$

Using Markov's inequality concludes the proof.

$\square$

**Lemma 4.** *For any $a, b > 0$, such that $1 + a + a\alpha > 2b$, there exists constants $C_{1,p}, C_{2,p}$, only depending on $\gamma_1$ and $p$, such that defining*

$$\lambda_{c,t} = \begin{cases} \rho_* \dfrac{1 - e^{-4C_\infty}}{4C_\infty}\,\varepsilon_t, & \text{if } c = 0, \\[3mm] \rho_*(1 - e^{-1})\,\varepsilon_t\, t^c, & \text{if } c \in (0, a]. \end{cases}$$

*we have for any $c \in [0, a]$*

$$\mathbb{E}[\Delta_{t+1}^p] \leq \mathbb{E}[\Delta_t^p]\left(1 - \gamma_{t+1}\lambda_{c,t} + C_{1,p}\gamma_{t+1}^2\right) + \gamma_{t+1}D_C^{2p}\mathbb{P}\left(\Delta_t \geq t^{-2c}\right)\mathbf{1}_{c\neq 0} + C_{2,p}\lambda_{c,t}^{-p+1}\gamma_{t+1}^{p+1}. \tag{18}$$

*Proof.* Let us fix $c \in [0, a]$. Starting from equation (13), raising to the power $p$ gives

$$\Delta_{t+1}^p \leq \left(\Delta_t - 2\gamma_{t+1}\langle\nabla_{\mathbf{g}}h_{\varepsilon_t}(\mathbf{g}_t, X_{t+1}), \mathbf{g}_t - \mathbf{g}_t^*\rangle + 5\gamma_{t+1}^2\right)^p. \tag{19}$$

We note $\binom{p}{i,j,k} = \frac{p!}{i!j!k!}$ and apply the trinomial expansion to obtain

$$
\begin{aligned}
\Delta_{t+1}^p &\leq \sum_{\substack{i,j,k \\ i+j+k=p}} \binom{p}{i,j,k} \Delta_t^i \left(-2\gamma_{t+1}\langle\nabla_{\mathbf{g}}h_{\varepsilon_t}(\mathbf{g}_t, X_{t+1}), \mathbf{g}_t - \mathbf{g}_t^*\rangle\right)^j 5^k \gamma_{t+1}^{2k} \\
&\leq \Delta_t^p - 2p\Delta_t^{p-1}\gamma_{t+1}\langle\nabla_{\mathbf{g}}h_{\varepsilon_t}(\mathbf{g}_t, X_{t+1}), \mathbf{g}_t - \mathbf{g}_t^*\rangle \\
&\quad - 2p(p-1)\Delta_t^{p-2}\gamma_{t+1}\langle\nabla_{\mathbf{g}}h_{\varepsilon_t}(\mathbf{g}_t, X_{t+1}), \mathbf{g}_t - \mathbf{g}_t^*\rangle 5\gamma_{t+1}^2 \\
&\quad + \sum_{\substack{i,j,k \\ i+j+k=p \\ (i,j,k)\notin\{(p,0,0),(p-1,1,0),(p-2,1,1)\}}} \binom{p}{i,j,k} \Delta_t^i \left(-2\gamma_{t+1}\langle\nabla_{\mathbf{g}}h_{\varepsilon_t}(\mathbf{g}_t, X_{t+1}), \mathbf{g}_t - \mathbf{g}_t^*\rangle\right)^j 5^k \gamma_{t+1}^{2k}.
\end{aligned}
$$

We divide the set

$$\mathcal{S} := \left\{(i,j,k) \in \mathbb{N}^3, i+j+k = p, (i,j,k) \notin (p,0,0),(p-1,1,0),(p-2,1,1)\right\}$$

into the following partition

$$
\begin{aligned}
\mathcal{P}_a &:= (i,j,k) \in \{(p-2,2,0),(p-3,3,0),(p-1,0,1),(0,p,0)\}, \\
\mathcal{P}_b &:= (\mathcal{S} \setminus \mathcal{P}_a) \cap \{i=0\}, \\
\mathcal{P}_c &:= (\mathcal{S} \setminus \mathcal{P}_a) \cap \{i \neq 0\} \cap \{j \geq 4\} \cap \{k=0\}, \\
\mathcal{P}_d &:= (\mathcal{S} \setminus \mathcal{P}_a) \cap \{i \neq 0\} \cap \{0 < j < 4\} \cap \{k \neq 0\}, \\
\mathcal{P}_e &:= (\mathcal{S} \setminus \mathcal{P}_a) \cap \{i \neq 0\} \cap \{j=0\} \cap \{k \neq 0\}.
\end{aligned}
$$

In what follows, the constants $C_k$ may depend on the constant $\gamma_1$ from the learning rate $\gamma_t = \gamma_1/t^b$, since we will often use the crude bound $\gamma_t^{p+1+k} \leq \gamma_1^k \gamma_t^{p+1}$ for $k \in \mathbb{N}$. Note, however, that $\lambda_{c,t} \leq 1$ for all $t$, and therefore $\lambda_{c,t}^{-q+k} \leq \lambda_{c,t}^{-q}$ for all $q, k \in \mathbb{N}$, so the constants $C_k$ will not depend on $\lambda_{c,t}$.

We also introduce, for all $p$, the constant

$$\Gamma_p = \frac{3p+1}{2p-1}.$$

**Case where $(i,j,k) \in \mathcal{P}_a$.**

If $(i,j,k) = (p-2,2,0)$:

$$
\begin{aligned}
\binom{p}{p-2,2,0} &\Delta_t^{p-2} \left(-2\gamma_{t+1}\langle\nabla_{\mathbf{g}}h_{\varepsilon_t}(\mathbf{g}_t, X_{t+1}), \mathbf{g}_t - \mathbf{g}_t^*\rangle\right)^2 \\
&\leq 8p(p-1)\Delta_t^{p-1}\gamma_{t+1}^2 && \text{by Cauchy-Schwarz} \\
&\leq 8p(p-1)\left(\Delta_t^p\frac{p-1}{p}c_1^{p/(p-1)} + \gamma_{t+1}^p\frac{1}{pc_1^p}\right)\gamma_{t+1} && \text{by Young: } q = \tfrac{p}{p-1}, q' = p, c_1 > 0 \\
&\leq p\Delta_t^p\frac{\gamma_{t+1}\lambda_{c,t}}{\Gamma_p} + \gamma_{t+1}^{p+1}\frac{1}{p}\left(\frac{8\Gamma_p(p-1)}{\lambda_{c,t}}\right)^{p-1} && \text{taking } c_1 = \left(\frac{\lambda_{c,t}}{8\Gamma_p(p-1)}\right)^{(p-1)/p} \\
&\leq p\Delta_t^p\frac{\gamma_{t+1}\lambda_{c,t}}{\Gamma_p} + C_1\lambda_{c,t}^{-p+1}\gamma_{t+1}^{p+1} && \text{defining } C_1 \text{ readily.}
\end{aligned}
$$

If $(i, j, k) = (p - 3, 3, 0)$:

$$\binom{p}{p-3, 3, 0} \Delta_t^{p-3} \left(-2\gamma_{t+1} \langle \nabla_{\mathbf{g}} h_{\varepsilon_t}(\mathbf{g}_t, X_{t+1}), \mathbf{g}_t - \mathbf{g}_t^* \rangle\right)^3$$

$$\leq 4^3 \Delta_t^{p-\frac{3}{2}} \gamma_{t+1}^3 \qquad \text{by Cauchy-Schwarz}$$

$$\leq 64 \left( \Delta_t^p \frac{p - \frac{3}{2} \binom{p}{p-3}^{2p/3}}{p} c_2^{p/(p-\frac{3}{2})} + \gamma_{t+1}^{4p/3} \frac{3}{2pc_2^{2p/3}} \right) \gamma_{t+1}$$

$$\qquad \text{by Young: } q = \frac{p}{p-3/2}, q' = \frac{2p}{3}$$

$$\leq p \Delta_t^p \frac{\gamma_{t+1} \lambda_{c,t}}{\Gamma_p} + \gamma_{t+1}^{\frac{4}{3}p+1} \frac{3}{2p} \left( \frac{32 \Gamma_p (p-1)(p-2) \binom{p}{p-3}}{3\lambda_{c,t}} \right)^{\frac{2p-3}{3}}$$

$$\qquad \text{taking } c_2 = \left( \frac{3\lambda_{c,t}}{32\Gamma_p(p-1)(p-2)} \right)^{\frac{p-\frac{3}{2}}{p}}$$

$$\leq p \Delta_t^p \frac{\gamma_{t+1} \lambda_{c,t}}{\Gamma_p} + C_2 \lambda_{c,t}^{-p+1} \gamma_{t+1}^{p+1} \qquad \text{since } \frac{4}{3}p + 1 \geq p + 2 \text{ and } \frac{2p-3}{3} \leq p - 1 .$$

If $(i, j, k) = (p - 1, 0, 1)$:

$$\binom{p}{p-1, 0, 1} \Delta_t^{p-1} \gamma_{t+1}^2 \leq p \left( \frac{p-1}{p} c_3^{\frac{p-1}{p}} \Delta_t^p + \frac{5^p}{pc_3^p} \gamma_{t+1}^p \right) \gamma_{t+1} \qquad \text{by Young: } q = \frac{p}{p-1}, q' = p$$

$$\leq p \Delta_t^p \frac{\gamma_{t+1} \lambda_{c,t}}{\Gamma_p} + \left( \frac{5^p \Gamma_p}{\lambda_{c,t}} \right)^p \gamma_{t+1}^{p+1} \qquad \text{taking } c_3 = \left( \frac{\lambda_{c,t}}{\Gamma_p} \right)^{\frac{p-1}{p}}$$

$$\leq p \Delta_t^p \frac{\gamma_{t+1} \lambda_{c,t}}{\Gamma_p} + C_3 \lambda_{c,t}^{-p+1} \gamma_{t+1}^{p+1} \qquad \text{defining } C_3 \text{ readily.}$$

If $(i, j, k) = (0, p, 0)$:

$$\binom{p}{0, p, 0} \left(-2\gamma_{t+1} \langle \nabla_{\mathbf{g}} h_{\varepsilon_t}(\mathbf{g}_t, X_{t+1}), \mathbf{g}_t - \mathbf{g}_t^* \rangle\right)^p$$

$$\leq 4^p \left( \frac{c_4^2}{2} \Delta_t^p + \frac{1}{2c_4^2} \gamma_{t+1}^{2p-2} \right) \gamma_{t+1} \qquad \text{Cauchy-Schwarz and Young : } q = q' = 2$$

$$\leq \Delta_t^p \frac{\gamma_{t+1} \lambda_{c,t}}{\Gamma_p} + \frac{1}{2} \left( \frac{4^{2p} \Gamma_p}{4\lambda_{c,t}} \right) \gamma_{t+1}^{2p-1} \qquad \text{taking } c_4 = \left( \frac{2\lambda_{c,t}}{4^p \Gamma_p} \right)^{\frac{1}{2}}$$

$$\leq \Delta_t^p \frac{\gamma_{t+1} \lambda_{c,t}}{\Gamma_p} + C_4 \lambda_{c,t}^{-1} \gamma_{t+1}^{p+1} \qquad \text{since } 2p - 1 \geq p + 1, \text{ and defining } C_4 \text{ readily.}$$

**Case where $(i, j, k) \in \mathcal{P}_b$.**

We have $j + k = p$ such that $j + 2k \geq p + 1$ since $k \neq 0$. Using the bound $\|\mathbf{g}_t - \mathbf{g}_{\varepsilon_t}^*\| \leq D_{\mathcal{C}}$, we obtain:

$$\sum_{(i,j,k) \in \mathcal{P}_b} \binom{p}{i, j, k} \left(-2\gamma_{t+1} \langle \nabla_{\mathbf{g}} h_{\varepsilon_t}(\mathbf{g}_t, X_{t+1}), \mathbf{g}_t - \mathbf{g}_t^* \rangle\right)^j 5^k \gamma_{t+1}^{2k}$$

$$\leq \sum_{(i,j,k) \in \mathcal{P}_b} \binom{p}{i, j, k} (4D)^j 5^k \gamma_{t+1}^{j+2k}$$

$$\leq C_5 \gamma_{t+1}^{p+1} \qquad \text{defining } C_5 \text{ readily.}$$

**Case where $(i, j, k) \in \mathcal{P}_c$.**

$$\sum_{(i,j,k)\in\mathcal{P}_c}\binom{p}{i,j}\Delta_t^i\left(-2\gamma_{t+1}\left\langle\nabla_{\mathbf{g}}h_{\varepsilon_t}(\mathbf{g}_t,X_{t+1}),\mathbf{g}_t-\mathbf{g}_t^*\right\rangle\right)^j \qquad\text{by Cauchy-Schwarz}$$

$$\le\sum_{(i,j,k)\in\mathcal{P}_c}\binom{p}{i,j}\Delta_t^{i+j/2}4^j\gamma_{t+1}^{j-2}\gamma_{t+1}^2 \qquad\text{by Young: }q=\frac{p}{i+j/2},q'=\frac{2p}{j}$$

$$\le\sum_{(i,j,k)\in\mathcal{P}_c}\binom{p}{i,j}\left(\frac{i+j/2}{p}\Delta_t^p+\frac{j}{2p}\gamma_{t+1}^{2p(j-2)/j}\right)\gamma_{t+1}^2$$

$$\le\sum_{(i,j,k)\in\mathcal{P}_c}\binom{p}{i,j}4^j\left(\Delta_t^p\gamma_{t+1}^2+\frac{1}{2}\gamma_{t+1}^{p+1}\right) \qquad j\ge 4\text{ so}:\frac{2p(j-2)}{j}+2\ge p+1$$

$$\le 8^p\Delta_t^p\gamma_{t+1}^2+8^p\gamma_{t+1}^{p+1}\ .$$

**Case where $(i,j,k)\in\mathcal{P}_d$.**

$$\sum_{(i,j,k)\in\mathcal{P}_d}\binom{p}{i,j}\Delta_t^i\left(-2\gamma_{t+1}\left\langle\nabla_{\mathbf{g}}h_{\varepsilon_t}(\mathbf{g}_t,X_{t+1}),\mathbf{g}_t-\mathbf{g}_t^*\right\rangle\right)^j5^k\gamma_{t+1}^{2k}$$

$$\le\sum_{(i,j,k)\in\mathcal{P}_d}\binom{p}{i,j,k}5^k\Delta_t^{i+j/2}4^j\gamma_{t+1}^{j+2k} \qquad\text{by Cauchy-Schwarz}$$

$$\le\sum_{(i,j,k)\in\mathcal{P}_d}\binom{p}{i,j,k}4^j\left(c_6^q\frac{i+j/2}{p}\Delta_t^p+c_6^{-q'}\frac{jC_\gamma^{2p}}{2p}\gamma_{t+1}^{2p}\right)$$

$$\text{by Young: }q=\frac{p}{i+j/2},q'=\frac{2p}{j+2k}\ .$$

Taking $c_6=\gamma_t^{2/q}$, it comes $c_6^{-q'}=\gamma_t^{-2q'/q}=\gamma_t^{-2/(q-1)}$. Since we are only considering cases with $i,j,k\ge 1$ (which forces $p\ge 3$) and we are excluding the particular case $(i,j,k)=(p-2,1,1)$, one can show that the parameter $q=\frac{2p}{2p-2i-j}=\frac{2p}{2k+j}$ satisfies

$$\frac{2p}{2p-4}\le q\le\frac{2p}{3}$$

$$\frac{4}{2p-4}\le q-1\le\frac{2p-3}{3}\ .$$

Thus, since $\frac{2}{q-1}\le p-2$, it follows that

$$2p-\frac{2}{q-1}\ge 2p-(p-2)=p+2\ge p+1\ .$$

Therefore, using the crude bound $\sum_{(i,j,k)\in\mathcal{P}_d}\binom{p}{i,j,k}\le 3^p$ and defining a constant $C_6$ readily, we obtain

$$\sum_{(i,j,k)\in\mathcal{P}_d}\binom{p}{i,j}\Delta_t^i\left(-2\gamma_{t+1}\left\langle\nabla_{\mathbf{g}}h_{\varepsilon_t}(\mathbf{g}_t,X_{t+1}),\mathbf{g}_t-\mathbf{g}_t^*\right\rangle\right)^j5^k\gamma_{t+1}^{2k}\le 3^p\gamma_{t+1}^2\Delta_t^p+C_6\gamma_{t+1}^{p+1}\ .$$

**Case where $(i,j,k)\in\mathcal{P}_e$.**

Since $j=0,i+k=p$, and $(p-1,0,1)\in\mathcal{P}_a$, we have $k\ge 2$. We use Young's inequality with $q=\frac{p}{i},q'=\frac{p}{k}$ to obtain

$$\sum_{(i,j,k)\in\mathcal{P}_e}\binom{p}{i,k}\Delta^i\gamma_{t+1}^{2k}5^k\le\sum_{(i,j,k)\in\mathcal{P}_e}\binom{p}{i,k}\left(\frac{i}{p}\Delta^p+\frac{k}{p}(\gamma_{t+1}5^{\frac{1}{2}})^{\frac{p(2k-2)}{k}}\right)\gamma_{t+1}^2$$

$$\le\sum_{(i,j,k)\in\mathcal{P}_e}\binom{p}{i,k}\left(\Delta^p\gamma_{t+1}^2+5^{p-\frac{p}{k}}\gamma_{t+1}^{2p-\frac{2p}{k}+2}\right)$$

$$\le 2^p\Delta^p\gamma_{t+1}^2+C_7\gamma_{t+1}^{p+1} \qquad\text{since }2p-\frac{2p}{k}+2\ge p+2\ .$$

**Summing up the inequalities**, we obtain

$$\Delta_{t+1}^p \leq \Delta_t^p$$
$$- 2p\Delta_t^{p-1}\gamma_{t+1}\langle \nabla_{\mathbf{g}} h_{\varepsilon_t}(\mathbf{g}_t, X_{t+1}), \mathbf{g}_t - \mathbf{g}_t^*\rangle$$
$$- 2p(p-1)\Delta_t^{p-1}\gamma_{t+1}\langle \nabla_{\mathbf{g}} h_{\varepsilon_t}(\mathbf{g}_t, X_{t+1}), \mathbf{g}_t - \mathbf{g}_t^*\rangle 5\gamma_{t+1}^2$$
$$+ \Delta_t^p\lambda_{c,t}\frac{3p+1}{\Gamma_p}\gamma_{t+1}$$
$$+ \Delta_t^p\gamma_{t+1}^2(8^p + 3^p + 2^p)$$
$$+ \gamma_{t+1}^{p+1}\left((C_1 + C_2 + C_3)\lambda_{c,t}^{-p+1} + C_4\lambda_{c,t}^{-1} + C_5 + 8^p + C_6 + C_7\right).$$

By convexity of $H_{\varepsilon_t}$, taking the conditional expectation gives

$$\mathbb{E}\left[-2p(p-1)\Delta_t^{p-1}\gamma_{t+1}\langle \nabla_{\mathbf{g}} h_{\varepsilon_t}(\mathbf{g}_t, X_{t+1}), \mathbf{g}_t - \mathbf{g}_t^*\rangle 5\gamma_{t+1}^2 \mid \mathcal{F}_t\right] \leq 0,$$

Applying Lemma 2, recalling that $\lambda_{c,t} = \rho_* \frac{1-e^{-1}}{2\sqrt{2}c_\infty}\varepsilon_t$ if $c = 0$, and $\lambda_{c,t} = \rho_* \frac{1-e^{-1}}{\sqrt{2}}\varepsilon_t t^c$ if $c \in (0, a]$, we have

$$\langle \nabla H_t(\mathbf{g}), \mathbf{g} - \mathbf{g}_t^*\rangle \geq \rho_* \frac{\varepsilon_t}{\sqrt{2}\|\mathbf{g} - \mathbf{g}_t^*\|_\infty}\Delta_t \geq \begin{cases} \lambda_{c,t}\Delta_t \text{ if } c = 0 \\ \lambda_{c,t}\Delta_t \mathbf{1}_{\Delta_t \leq t^{-2c}} \text{ if } c \in (0, a]. \end{cases} \tag{20}$$

Therefore,

$$\mathbb{E}\left[-2p\Delta_t^{p-1}\gamma_{t+1}\langle \nabla_{\mathbf{g}} h_{\varepsilon_t}(\mathbf{g}_t, X_{t+1}), \mathbf{g}_t - \mathbf{g}_t^*\rangle \mid \mathcal{F}_t\right]$$
$$= -2p\Delta_t^{p-1}\gamma_{t+1}\mathbb{E}\left[\langle \nabla_{\mathbf{g}} h_{\varepsilon_t}(\mathbf{g}_t, X_{t+1}), \mathbf{g}_t - \mathbf{g}_t^*\rangle \mid \mathcal{F}_t\right]$$
$$= -2p\Delta_t^{p-1}\gamma_{t+1}\langle \nabla H_{\varepsilon_t}(\mathbf{g}_t), \mathbf{g}_t - \mathbf{g}_t^*\rangle \qquad\qquad \text{by (20)}$$
$$\leq -2p\lambda_{c,t}\gamma_{t+1}\Delta_t^p + \gamma_{t+1}D_{\mathcal{C}}^{2p}\mathbf{1}_{\Delta_t \geq t^{-2c}}\mathbf{1}_{c \neq 0} \qquad \text{using that } \Delta_t^p \leq D_{\mathcal{C}}^{2p}.$$

We now just have to sum up the inequalities. Fixing $\Gamma_p = \frac{3p+1}{2p-1}$ such that $-2p + \frac{3p+1}{\Gamma_p} = -1$, $C_{1,p} = 8^p + 3^p + 2^p$, $C_{2,p} = 8^p + \sum_{k=1}^7 C_k$, and taking the expectation, we have the desired form

$$\mathbb{E}[\Delta_{t+1}^p] \leq \mathbb{E}[\Delta_t^p]\left(1 - \gamma_{t+1}\lambda_{c,t} + C_{1,p}\gamma_{t+1}^2\right) + \gamma_{t+1}D_{\mathcal{C}}^{2p}\mathbb{E}\left[\mathbf{1}_{\Delta_t \geq t^{-2c}}\right]\mathbf{1}_{c \neq 0} + C_{2,p}\lambda_{c,t}^{-p+1}\gamma_{t+1}^{p+1}.$$

$\square$

## B.6 Proof of Lemma 1: Projection step

*Proof.* According to [40], any optimal pair of functions $(f_\varepsilon, g_\varepsilon)$ solving the dual formulation of entropic OT with regularization $\varepsilon \geq 0$ satisfies the Schrödinger equations. That is, we can take for all $y \in \mathbb{R}^d$, $g_\varepsilon(y) = f_\varepsilon^{c,\varepsilon}(y)$. Moreover, $\frac{1}{2}\|x - y\|^2$ is $R$-Lipschitz on $B(0, R)$. Therefore, since by Assumption 1, we have $\text{Supp}(\mu) \subset B(0, R)$ and $\text{Supp}(\nu) \subset B(0, R)$, we can exploit the Lipschitz property of our cost function on $B(0, R)$. Using that the $(c, \varepsilon)$-transform has the same modulus of continuity as $c$ (see Lemma 3.1 in [40]), we get, for all $y, y' \in \mathbb{R}^d$:

$$|f_\varepsilon^{c,\varepsilon}(y) - f_\varepsilon^{c,\varepsilon}(y')| \leq R\|y - y'\|.$$

That is, coming back to the function $g$, we have for all $j, j' \in [\![1, M]\!]$ :

$$|g_\varepsilon(y_j) - g_\varepsilon(y_{j'})| \leq R\|y_j - y_{j'}\|.$$

By writing back our dual potential as a vector, that is $\mathbf{g}^* = (g_1^*, \ldots, g_M^*)$, where for all $j \in [\![1, M]\!]$, $g_j^* = g_\varepsilon(y_j)$, we have

$$|g_j^* - g_{j'}^*| \leq R\|y_j - y_{j'}\|.$$

Moreover, if $\mathbf{g}^*$ optimizes the semi-dual $H_\varepsilon$, then for any $\beta \in \mathbb{R}$, the vector $\mathbf{g}^* + \beta\mathbf{1}_M$ optimizes $H_\varepsilon$. In particular, $\mathbf{g}^* - g_\varepsilon(y_1)\mathbf{1}_M$, which we rename $\mathbf{g}^*$, optimizes the semi-dual, with $g_1^* = 0$. Hence, for all $j \in 1, \ldots, M$

$$\left|g_{y_1}^* - g_{y_j}^*\right| = \left|g_{y_j}^*\right| \leq R\|y_1 - y_j\|.$$

That is, there exists an optimizer in the desired closed convex set. $\square$

**Remark:** Note that for other costs such as $c(x, y) = \|x - y\|$ which defines the 1-Wasserstein distance, this projection set can be more relevant. Indeed, in this case, the cost is 1-Lipschitz and the projection set depends only on the target measure $\nu$ and no assumption of bounded cost is needed. In this case, the practitioner could choose the index $k$ such that $g_k = 0$, minimizing for instance the Euclidean diameter of the corresponding set.

### B.7 Proof of Lemma 2: Global and local RSC condition of $H_\varepsilon$

*Proof.* For any $\mathbf{g} \in \mathcal{C}$ and $s \in [0, 1]$, note $\mathbf{g}_s = \mathbf{g}_\varepsilon^* + s(\mathbf{g} - \mathbf{g}_\varepsilon^*)$, where $\mathbf{g}_\varepsilon^*$ is the minimizer of $H_\varepsilon$ satisfying $\sum_{i=1}^M g_i = \sum_{i=1}^M g_{\varepsilon,i}^*$ and define $\varphi$ by

$$\varphi : s \in [0, 1] \mapsto H_\varepsilon(\mathbf{g}_s) .$$

Applying Lemma 5, whose proof is postponed until after this one, we have that

$$|\varphi'''(s)| \leq \frac{1}{\varepsilon} \varphi''(s) \max_{1 \leq j \leq M} |g_j - g_j^* - m(x, \mathbf{g}_s)| , \tag{21}$$

where for all $x \in \mathbb{R}^d : m(x, \mathbf{g}_s) := \sum_{j=1}^M \chi_j^\varepsilon(x, \mathbf{g}_s)(\mathbf{g}_s - \mathbf{g}_\varepsilon^*)$.

Using Hölder's inequality with the Hölder conjugates $p = 1, q = +\infty$ for $\delta_0$ and Cauchy-Schwarz inequality as in [5] for $\delta_1$, we obtain

$$\frac{1}{\varepsilon} \max_{1 \leq j \leq M} |g_j - g_{\varepsilon,j}^* - m(x, \mathbf{g}_s)| \leq \begin{cases} \frac{2}{\varepsilon}\|\mathbf{g} - \mathbf{g}_\varepsilon^*\|_\infty =: \delta_0 , \\ \frac{\sqrt{2}}{\varepsilon}\|\mathbf{g} - \mathbf{g}_\varepsilon^*\| =: \delta_1 . \end{cases} \tag{22}$$

Use $\delta = \delta_0$ or $\delta_1 = 1$. Since $\sum_{i=1}^M g_i = \sum_{i=1}^M g_{\varepsilon,i}^*$, $\varphi$ is strictly convex, and therefore, we can divide by $\varphi''(s)$ to obtain for $s \in [0, 1]$

$$\frac{\varphi'''(s)}{\varphi''(s)} \geq -\delta .$$

Integrating between 0 and $S$ and using that $\int_0^S \frac{\varphi'''(s)}{\varphi''(s)} ds = \ln|\varphi''(S)| - \ln|\varphi''(0)|$ gives

$$\varphi''(s) \geq \exp(-\delta S)\varphi''(0) . \tag{23}$$

Since $\varphi''(s) = (\mathbf{g} - \mathbf{g}_\varepsilon^*)^\top \nabla^2 H_\varepsilon(\mathbf{g}_s)(\mathbf{g} - \mathbf{g}_\varepsilon^*)$, recalling that $\rho^*$ is the second smallest eigenvalue of $\nabla^2 H_\varepsilon(\mathbf{g}_\varepsilon^*)$ gives the upper bound

$$\varphi''(0) \geq \rho^* \|\mathbf{g} - \mathbf{g}_\varepsilon^*\|^2 .$$

Then, since $\varphi'(s) = \langle \nabla H_\varepsilon(\mathbf{g}_s), \mathbf{g} - \mathbf{g}_\varepsilon^* \rangle$, an integration of (23) between 0 and 1 gives

$$\langle \nabla H_\varepsilon(\mathbf{g}), \mathbf{g} - \mathbf{g}_\varepsilon^* \rangle_v \geq \rho^* \frac{1}{\delta} (1 - \exp(-\delta)) \|\mathbf{g} - \mathbf{g}_\varepsilon^*\|^2. \tag{24}$$

Note that the function $\delta \in (0, \infty) \mapsto \frac{1}{\delta}(1 - \exp(-\delta))$ is strictly decreasing and upper bounded by 1. If $\varepsilon = 1$, take $\delta = \delta_0 = 2\|\mathbf{g} - \mathbf{g}_\varepsilon\|_\infty$ and use the fact that $\|\mathbf{g} - \mathbf{g}_\varepsilon\|_\infty \leq 2C_\infty$ to obtain

$$\langle \nabla H_1(\mathbf{g}), \mathbf{g} - \mathbf{g}_1^* \rangle \geq \rho^* \frac{1}{4C_\infty} \left(1 - e^{-4C_\infty}\right) \|\mathbf{g} - \mathbf{g}_\varepsilon^*\|^2 .$$

In the same way, for $\varepsilon < 1$ and $\|\mathbf{g} - \mathbf{g}_\varepsilon^*\| \leq \frac{\varepsilon}{\sqrt{2}}$, taking $\delta = \delta_1 = \sqrt{2}\|\mathbf{g} - \mathbf{g}_\varepsilon^*\| \leq 1$ we obtain

$$\langle \nabla H_\varepsilon(\mathbf{g}), \mathbf{g} - \mathbf{g}_\varepsilon^* \rangle_v \geq \rho^* \left(1 - e^{-1}\right) \|\mathbf{g} - \mathbf{g}_\varepsilon^*\|^2 ,$$

which concludes the proof.

$\square$

**Lemma 5** (Helping Lemma for the RSC condition of $H_\varepsilon$). *For any* $\mathbf{g} \in \mathcal{C}$ *and* $t \in [0, 1]$, *define* $\mathbf{g}_s = \mathbf{g}_\varepsilon^* + s(\mathbf{g} - \mathbf{g}_\varepsilon^*)$, *where* $\mathbf{g}_\varepsilon^*$ *is the minimizer of* $H_\varepsilon$ *satisfying* $\sum_{i=1}^M g_i = \sum_{i=1}^M g_{\varepsilon,i}^*$. *The function* $\varphi$, *defined by*

$$\varphi : s \in [0, 1] \mapsto H_\varepsilon(\mathbf{g}_s) ,$$

*satisfies*

$$|\varphi'''(s)| \leq \frac{1}{\varepsilon}\varphi''(s) \max_{1 \leq j \leq M} \left|g_j - g_j^* - m\left(x, \mathbf{g}_s\right)\right| .$$

*Where* $m(x, \mathbf{g}_s) = \sum_{i=1}^{M} \chi_i^\varepsilon(x, \mathbf{g}_s)(g_i - g_{\varepsilon,i}^*).$

*Proof.* The proof is an adaptation of the proof of Lemma A.2 in [5]. For completeness, we recall all the steps of their proof that are needed for our results. Note that their recent erratum regarding Lemma A.1 has no impact on Lemma A.2.

For any $\mathbf{g} \in \mathbb{R}^M$ and $s \in [0, 1]$, define $\mathbf{g}_s = \mathbf{g}_\varepsilon^* + s(\mathbf{g} - \mathbf{g}_\varepsilon^*)$, where $\mathbf{g}_\varepsilon^*$ is the minimizer of $H_\varepsilon$ satisfying $\sum_{i=1}^M g_i = \sum_{i=1}^M g_{\varepsilon,i}^*$. We also define the function $\varphi$ by

$$\varphi : s \in [0, 1] \mapsto H_\varepsilon(\mathbf{g}_s) .$$

Its first to third-order derivatives are given by

$$\varphi'(s) = \langle \nabla H_\varepsilon(\mathbf{g}_s), \mathbf{g} - \mathbf{g}_\varepsilon^* \rangle ,$$
$$\varphi''(s) = (\mathbf{g} - \mathbf{g}_\varepsilon^*)^\top \nabla^2 H_\varepsilon(\mathbf{g}_s)(\mathbf{g} - \mathbf{g}_\varepsilon^*) ,$$
$$\varphi'''(s) = \sum_{i,j,k=1}^{M} \frac{\partial^3 H_\varepsilon(\mathbf{g}_s)}{\partial g_i \partial g_j \partial g_k}(\mathbf{g} - \mathbf{g}_\varepsilon^*)_i \, (\mathbf{g} - \mathbf{g}_\varepsilon^*)_j \, (\mathbf{g} - \mathbf{g}_\varepsilon^*)_k .$$

Since for all $\mathbf{g} \in \mathbb{R}^M$, $\nabla H_\varepsilon(\mathbf{g}) = -\mathbb{E}_{X \sim \mu}\left[\chi^\varepsilon(X, \mathbf{g})\right] + \mathbf{w}$,

$$\varphi'(s) = \langle -\mathbb{E}_{X \sim \mu}\left[\chi^\varepsilon(X, \mathbf{g}_s)\right] + \mathbf{w}, \mathbf{g} - \mathbf{g}_\varepsilon^* \rangle$$
$$= -\mathbb{E}_{X \sim \mu}\left[m(X, \mathbf{g}_s)\right] + \langle \mathbf{w}, \mathbf{g} - \mathbf{g}_\varepsilon^* \rangle ,$$

defining for all $x \in \mathbb{R}^d$, $m(x, \mathbf{g}_s) = \sum_{i=1}^M \chi_i^\varepsilon(x, \mathbf{g}_s)(g_i - g_{\varepsilon,i}^*) .$

Using that $\nabla_{\mathbf{g}} \chi^\varepsilon(x, \mathbf{g}) = \frac{1}{\varepsilon}\left(\mathrm{diag}(\chi^\varepsilon(x, \mathbf{g})) - \chi^\varepsilon(x, \mathbf{g})\chi^\varepsilon(x, \mathbf{g})^\perp\right)$, we have

$$\frac{d}{ds}\chi^\varepsilon(X, \mathbf{g}_s) = \frac{1}{\varepsilon}\left(\mathrm{diag}(\chi^\varepsilon(X, \mathbf{g}_s)) - \chi^\varepsilon(X, \mathbf{g}_s)\chi^\varepsilon(X, \mathbf{g}_s)^\top\right)(\mathbf{g} - \mathbf{g}_\varepsilon^*).$$

Therefore, using the expression of $m$ yields to

$$\varphi''(s) = -\frac{1}{\varepsilon}\mathbb{E}_{X \sim \mu}\left[(\mathbf{g} - \mathbf{g}_\varepsilon^*)^\top \mathrm{diag}(\chi_i^\varepsilon(X, \mathbf{g}_s))(\mathbf{g} - \mathbf{g}_\varepsilon^*) - m(X, \mathbf{g}_s)^2\right]$$
$$= -\frac{1}{\varepsilon}\mathbb{E}_{X \sim \mu}\left[\sigma^2(X, \mathbf{g}_s)\right]$$

defining for all $x \in \mathbb{R}^d$

$$\sigma^2(x, \mathbf{g}_s) = \sum_{i=1}^{M} \chi_i^\varepsilon(x, \mathbf{g}_s)(g_i - g_{\varepsilon,i}^*)^2 - (m(x, \mathbf{g}_s))^2$$
$$= \sum_{i=1}^{M} \chi_i^\varepsilon(x, \mathbf{g}_s)\left(g_i - g_{\varepsilon,i}^* - m(x, \mathbf{g}_s)\right)^2 .$$

A derivation of $\sigma^2$ leads to (see [5] eq. (A.19),) for more details)

$$-\varepsilon\frac{d}{ds}\sigma^2(x, \mathbf{g}_s) = \sum_{i=1}^{M} \chi_i(x, \mathbf{g}_s)(g_i - g_{\varepsilon,i}^*)^3 - 3m(x, \mathbf{g}_s)\sigma^2(x, \mathbf{g}_s) - (m(x, \mathbf{g}_s))^3$$
$$= \sum_{i=1}^{M} \chi_i^\varepsilon(x, \mathbf{g}_s)\left(g_i - g_{\varepsilon,i}^* - m(x, \mathbf{g}_s)\right)^3 .$$

Since $\varepsilon^2\varphi'''(s) = \mathbb{E}_{X \sim \mu}\left[\frac{d}{ds}\sigma^2(X, \mathbf{g}_s)\right]$, we conclude

$$|\varphi'''(s)| \leq \frac{1}{\varepsilon}\varphi''(s) \max_{1 \leq j \leq M} \left|g_j - g_j^* - m\left(x, \mathbf{g}_s\right)\right| .$$

$\square$

## B.8 Other Technical results

**Proposition 4.** *For all* $\mathbf{g} \in \mathcal{C}$ *and all* $\alpha' \in (0, \alpha)$, *we have*

$$\|\nabla H_\varepsilon(\mathbf{g}) - \nabla H_0(\mathbf{g})\|_\infty \lesssim \varepsilon^{1+\alpha'} .$$

*Proof.* We adopt the decomposition of $\mathcal{X}$ from Appendix A.3 of [18]. See Figure 1 in [18] for an illustration of this decomposition. Fix $i \in \{1, \ldots, M\}$, let $\mathcal{X} \subset B(0, R)$ be the support of $\mu$, and choose parameters

$$\eta = \varepsilon^\beta, \qquad \gamma = \tfrac{1}{2}\eta, \qquad \beta \in (0, 1).$$

Define, for all $i \in [\![1, M]\!]$, the function

$$f_i(x) := -c(x, y_i) + g_i = -\frac{1}{2}\|x - y_i\|^2 + g_i,$$

and use these to define the following sets:

$$H_{ij} = \mathbb{L}_i(\mathbf{g}) \cap \mathbb{L}_j(\mathbf{g})$$

$$\mathcal{X}_{i,\eta,+} = \left\{ x \in \mathbb{L}_i(\mathbf{g}); \forall j \neq i, \ \frac{f_i(x) - f_j(x)}{\|y_i - y_j\|} \geq \eta \right\},$$

$$\mathcal{X}_{i,\eta,-} = \left\{ x \in \mathbb{R}^d; \arg\max_j f_j(x) = k, \ \frac{f_k(x) - f_i(x)}{\|y_k - y_i\|} \geq \eta \right\},$$

$$H_{ij}^\gamma = \{ x \in H_{ij} \mid \forall k \notin \{i, j\}, \ f_i(x) = f_j(x) \geq f_k(x) + \gamma \max(\|y_i - y_k\|, \|y_j - y_k\|) \},$$

$$A_{i,\eta,\gamma} = \bigcup_{j \neq i} \{ x + t\, d_{ij} \ ; x \in H_{ij}^\gamma, \ t \in [-\eta\|y_i - y_j\|, \eta\|y_i - y_j\|] \}, \qquad d_{ij} = \frac{y_i - y_j}{\|y_i - y_j\|},$$

$$B_{i,\eta,\gamma} = \mathbb{R}^d \setminus (\mathcal{X}_{i,\eta,+} \cup \mathcal{X}_{i,\eta,-} \cup A_{i,\eta,\gamma}).$$

We also recall the point-wise definitions of the regularized and non-regularized functions constituting the gradients

$$\chi_i^\varepsilon(x) = \frac{\exp(f_i(x)/\varepsilon)}{\sum_{k=1}^M \exp(f_k(x)/\varepsilon)}, \qquad \chi_i(x) = \mathbf{1}\{ i = \arg\max_k f_k(x) \}.$$

and define the constant

$$c_y = \min_{i \neq j} \|y_i - y_j\| > 0 .$$

**Error decomposition.**

$$\|\nabla H_\varepsilon(\mathbf{g}) - \nabla H_0(\mathbf{g})\|_\infty = \max_i \left| \int_{\mathcal{X}} (\chi_i^\varepsilon - \chi_i) \, d\mu \right| \leq I_1 + I_2 + I_3 + I_4,$$

with

$$I_1 = \int_{\mathcal{X}_{i,\eta,+}} |\chi_i^\varepsilon - \chi_i|\, d\mu, \ I_2 = \int_{\mathcal{X}_{i,\eta,-}} |\chi_i^\varepsilon - \chi_i|\, d\mu, \ I_3 = \int_{A_{i,\eta,\gamma}} |\chi_i^\varepsilon - \chi_i|\, d\mu, \ I_4 = \int_{B_{i,\eta,\gamma}} |\chi_i^\varepsilon - \chi_i|\, d\mu.$$

**1. Interior regions $\mathcal{X}_{i,\eta,+}$ and $\mathcal{X}_{i,\eta,-}$.**

For $x \in \mathcal{X}_{i,\eta,+}$ one has $f_i - f_j \geq \eta\|y_i - y_j\| \geq \eta c_y$ for every $j \neq i$, hence

$$|\chi_i(x) - \chi_i^\varepsilon(x)| = 1 - \chi_i^\varepsilon(x)$$

$$= 1 - \frac{e^{f_i/\varepsilon}}{\sum_{k=1}^M e^{f_k/\varepsilon}}$$

$$= \frac{\sum_{k \neq i} e^{f_k/\varepsilon}}{\sum_{k=1}^M e^{f_k/\varepsilon}}$$

$$\leq \mathcal{O}\left( e^{-\eta c_y/\varepsilon} \right)$$

In a same way, we obtain the same bound if $x \in \mathcal{X}_{i,\eta,-}$, therefore

$$I_1 + I_2 = \mathcal{O}\left(e^{-\eta c_y/\varepsilon}\right).$$

## 2. Simple slabs $A_{i,\eta,\gamma}$.

Inside one slab $T_{ij} = \left\{x + t\, d_{ij} \mid x \in H_{ij}^\gamma,\ t \in [-\eta\|y_i - y_j\|,\ \eta\|y_i - y_j\|]\right\}$, and we have $f_i(x) - f_j(x) = t\|y_i - y_j\|$. for a certain $t \in [-\eta\|y_i - y_j\|,\ \eta\|y_i - y_j\|]$. All other indices satisfy $f_k(x) - f_i(x) \leq -c_y\gamma$, so

$$\sum_{k \notin \{i,j\}} e^{(f_k - f_i)/\varepsilon} \leq (M-2)e^{-c_y\gamma/\varepsilon}.$$

Hence

$$\chi_i^\varepsilon(x) = \frac{1}{1 + e^{-t\|y_i-y_j\|/\varepsilon} + \sum_{k \notin \{i,j\}} e^{(f_k(x)-f_i(x))/\varepsilon}} = p_{\tilde\varepsilon}(t) + \mathcal{O}\left(e^{-\gamma c_y/\varepsilon}\right),$$

with $p_{\tilde\varepsilon}(t) = \left(1 + e^{-t/\tilde\varepsilon}\right)^{-1}$, $\tilde\varepsilon = \varepsilon/\|y_i - y_j\|$.

Introduce coordinates $x = z + t n_{ij}$ with $n_{ij} = \frac{y_i - y_j}{\|y_i - y_j\|}$ and $z \in H_{ij}$; the Jacobian of this change of coordinate is 1. Since $f_\mu$ is $\alpha$-Hölder, there exists $L > 0$ such that we can write $f_\mu(z + t n_{ij}) = f_\mu(z) + r_\alpha(z,t)$ with $|r_\alpha(z,t)| \leq L|t|^\alpha$. Note that the function $p_{\tilde\varepsilon}(t) - \mathbf{1}_{t>0}$ is odd. Writing $\sigma$ the Hausdorff measure of dimension $d - 1$, we obtain

$$\left| \int_{T_{ij}^{\eta,\gamma}} (\chi_i^\varepsilon - \chi_i)\, d\mu \right|$$

$$= \left| \int_{H_{ij}^\gamma \cap B(0,R)} \int_{-\eta\|y_i-y_j\|}^{\eta\|y_i-y_j\|} \left[ p_{\tilde\varepsilon}(t) - \mathbf{1}_{t>0} + \mathcal{O}(e^{-\gamma/\varepsilon}) \right] (f_\mu(z) + r_\alpha(z,t))\, dt\, d\sigma(z) \right|$$

$$= \left| \int_{H_{ij}^\gamma \cap B(0,R)} \int_{-\eta\|y_i-y_j\|}^{\eta\|y_i-y_j\|} \left[ p_{\tilde\varepsilon}(t) - \mathbf{1}_{t>0} \right] r_\alpha(z,t)\, dt\, d\sigma(z) \right.$$

$$+ \int_{H_{ij}^\gamma \cap B(0,R)} \int_{-\eta\|y_i-y_j\|}^{\eta\|y_i-y_j\|} \left[ p_{\tilde\varepsilon}(t) - \mathbf{1}_{t>0} \right] f_\mu(z)\, dt\, d\sigma(z)$$

$$+ \left. \int_{H_{ij}^\gamma \cap B(0,R)} \int_{-\eta\|y_i-y_j\|}^{\eta\|y_i-y_j\|} \mathcal{O}(e^{-\gamma/\varepsilon}) (f_\mu(z) + r_\alpha(z,t))\, dt\, d\sigma(z) \right|$$

$$\lesssim \mathrm{vol}_{d-1}\left(H_{ij}^\gamma \cap B(0,R)\right) \int_{-\eta\|y_i-y_j\|}^{\eta\|y_i-y_j\|} |t|^\alpha\, dt + \mathcal{O}\left(\eta\, e^{-\gamma c_y/\varepsilon}\right)$$

$$= \mathcal{O}\left(\eta^{1+\alpha}\right) + \mathcal{O}\left(\eta\, e^{-\gamma c_y/\varepsilon}\right).$$

Summing over $j \neq i$ yields

$$I_3 = \mathcal{O}(\eta^{1+\alpha}) + \mathcal{O}(\eta\, e^{-\gamma c_y/\varepsilon}).$$

## 3. Corner set $B_{i,\eta,\gamma}$.

As shown in [18], denoting by $\theta$ the maximum angle that can be formed by three non-aligned points of the target measure, each corner that constitutes $B_{i,\eta,\gamma}$ is included in a cylinder of volume at most $\frac{4\pi \operatorname{diam}(B(0,R))^{d-2}}{\cos(\theta/2)^2}\gamma^2$. Moreover, there are at most $M^2$ such corners. Therefore, $\mu(B_{i,\eta,\gamma}) = \mathcal{O}(\gamma^2) = \mathcal{O}(\eta^2)$, and so

$$I_4 = \mathcal{O}(\eta^2).$$

## 4. Choice of the exponent $\beta$.

Let $\alpha' \in (0, \alpha)$ and pick
$$\beta \in \left(\tfrac{1+\alpha'}{1+\alpha}, 1\right).$$
Then $\eta^{1+\alpha} = \varepsilon^{\beta(1+\alpha)} \leq \varepsilon^{1+\alpha'}$ and $\eta^2 = \varepsilon^{2\beta} \leq \varepsilon^{1+\alpha'}$. Exponential terms are even smaller, hence
$$\|\nabla H_\varepsilon(\mathbf{g}) - \nabla H_0(\mathbf{g})\|_\infty = \mathcal{O}\left(\varepsilon^{1+\alpha'}\right).$$
$\square$

**Proposition 5.** *Let $(\gamma_t)_{t \geq 0}$ and $(\nu_t)_{t \geq 0}$ be some positive and decreasing sequences and let $(\delta_t)_{t \geq 0}$, satisfying the following:*

- *The sequence $\delta_t$ follows the recursive relation:*
$$\delta_{t+1} \leq \left(1 - 2\omega\gamma_{t+1} + \eta\gamma_{t+1}^2\right)\delta_t + \nu_{t+1}\gamma_{t+1}, \tag{25}$$
  *with $\delta_0 \geq 0$ and $\omega, \eta > 0$.*

- *Let $\gamma_t$ converge to $0$.*

- *Let $t_0 = \inf\{t \geq 1 : \omega\gamma_{t+1} \leq 1; \; \eta\gamma_t \leq \omega\}$.*

*Then, for all $t \geq t_0$, we have the upper bound:*
$$\delta_t \leq \exp\left(-\omega \sum_{i=t_0+1}^{t} \gamma_i\right)\left(\sum_{k=t_0}^{t} \gamma_k\nu_k + \delta_{t_0}\right) + \frac{1}{\omega}\nu_{\lceil\frac{t}{2}\rceil-1} \leq \frac{1}{\omega}\nu_{\lceil\frac{t}{2}\rceil-1} + o(\nu_t).$$

*Proof.* For all $t \geq t_0$, since $1 - 2\omega\gamma_{t+1} + \eta\gamma_{t+1}^2 \leq 1 - \omega\gamma_{t+1}$, one has
$$\delta_t \leq \left(1 - \omega\gamma_{t+1} + \eta\gamma_{t+1}^2\right)\delta_t + \nu_{t+1}\gamma_{t+1}$$
$$\leq \underbrace{\prod_{i=t_0+1}^{t}(1 - \omega\gamma_i)\,\delta_{t_0}}_{=:U_{1,t}} + \underbrace{\sum_{k=t_0+1}^{t}\prod_{i=k+1}^{t}(1 - \omega\gamma_i)\,\gamma_k\nu_k}_{=:U_{2,t}}$$

One can consider two cases: $\lceil t/2 \rceil - 1 \leq t_0$ and $\lceil t/2 \rceil - 1 > t_0$.

**Case where $\lceil t/2 \rceil - 1 \leq t_0 < t$:** Since $\nu_k$ is decreasing,
$$U_{2,t} \leq \nu_{t_0+1}\sum_{k=t_0+1}^{t}\prod_{i=k+1}^{t}(1 - \omega\gamma_i)\,\gamma_k$$
$$= \frac{1}{\omega}\nu_{t_0+1}\sum_{k=t_0+1}^{t}\prod_{i=k+1}^{t}(1 - \omega\gamma_i) - \prod_{i=k}^{t}(1 - \omega\gamma_i)$$
$$= \frac{1}{\omega}\nu_{t_0+1}\left(1 - \prod_{i=t_0+1}^{t}(1 - \omega\gamma_i)\right)$$
$$\leq \frac{1}{\omega}\nu_{t_0+1}$$

Since $\nu_k$ is decreasing, it comes $U_{2,t} \leq \frac{1}{\omega}\nu_{\lceil t/2 \rceil}$.

**Case where $\lceil t/2 \rceil - 1 > t_0$:** As in [3], for all $m = t_0 + 1, \ldots, t$, one has
$$U_{2,t} \leq \exp\left(-\omega\sum_{k=m+1}^{t}\gamma_k\right)\sum_{k=t_0+1}^{m}\gamma_k\nu_k + \frac{1}{\omega}\nu_m$$

Then, taking $m = \lceil t/2 \rceil - 1$, it comes
$$U_{2,t} \leq \exp\left(-\omega\sum_{k=\lceil t/2 \rceil}^{t}\gamma_k\right)\sum_{k=t_0+1}^{\lceil t/2 \rceil-1}\gamma_k\nu_k + \frac{1}{\omega}\nu_{\lceil t/2 \rceil-1}$$

$\square$

**Corollary 2.** *Let $(\gamma_t)_{t\geq 0}$ and $(\nu_t)_{t\geq 0}$ be some positive and decreasing sequences and let $(\delta_t)_{t\geq 0}$ be a sequence satisfying the following:*

- *The sequence $\delta_t$ follows the recursive relation:*

$$\delta_{t+1} \leq \left(1 - 2\omega\gamma_{t+1} + \eta\gamma_{t+1}^2\right)\delta_t + \nu_{t+1}\gamma_{t+1}, \tag{26}$$

  *with $\delta_0 \geq 0$ and $\omega, \eta > 0$.*

- *Let $\gamma_t = c_\gamma t^{-\alpha}$ with $\alpha \in (0,1)$.*

- *Let $t_0 = \inf\left\{t \geq 1 : \omega\gamma_{t+1} \leq 1;\ \eta\gamma_t \leq \omega\right\}$.*

*Then, for all $t \in \mathbb{N}$, we have the upper bound:*

$$\delta_t \leq \frac{1}{\omega}\nu_{\frac{t}{2}-1} + o(\nu_t).$$

*Proof.* Applying Proposition 5, for all $t \geq t_0$, we have the upper bound:

$$\delta_t \leq \exp\left(-\omega\sum_{i=t_0+1}^{t}\gamma_i\right)\left(\sum_{k=t_0}^{t}\gamma_k\nu_k + \delta_{t_0}\right) + \frac{1}{\omega}\nu_{\lceil\frac{t}{2}\rceil-1}.$$

Approximating the sum $\sum_{s=t_0}^{t}\gamma_s$ via a Riemann sum lower bound for the function $x \mapsto \frac{1}{x^\alpha}$, and applying the logarithmic inequality $\log(1-x) \leq -x$, one can now bound $\prod_{i=t_0+1}^{t}(1-\omega\gamma_i)\delta_{t_0}$ as

$$\prod_{i=t_0+1}^{t}(1-\omega\gamma_i)\delta_{t_0} \leq \exp\left(-\omega\frac{c_\gamma}{1-\alpha}\left((t+1)^{1-\alpha} - (t_0+1)^{1-\alpha}\right)\right)\gamma_{t_0}\nu_{t_0}$$

$$\leq \exp\left(-\frac{\omega c_\gamma}{2}\left((t+1)^{1-\alpha} - (t_0+1)^{1-\alpha}\right)\right)\gamma_{t_0}\nu_{t_0}.$$

In a same way, since

$$\exp\left(-\omega\sum_{k=\lceil t/2\rceil}^{t}\gamma_k\right) \leq \exp\left(-\frac{\omega c_\gamma}{2}(t+1)^{1-\alpha}\right),$$

we obtain

$$\delta_t \leq \exp\left(-\frac{1}{2}\omega c_\gamma t^{1-\alpha}\right)\exp\left(\frac{1}{2}\omega c_\gamma (t_0+1)^{1-\alpha}\right)\left(\sum_{k=t_0}^{t}\gamma_k\nu_k + \delta_{t_0}\right) + \frac{1}{\omega}\nu_{\frac{t}{2}-1}.$$

Since the product involving exponential terms converges exponentially fast, we finally obtain the desired convergence rate

$$\delta_t \leq \frac{1}{\omega}\nu_{\frac{t}{2}-1} + o(\nu_t).$$

$\square$

