# OpenReview forum: "Decreasing Entropic Regularization Averaged Gradient for Semi-Discrete Optimal Transport"
_NeurIPS.cc/2025/Conference — NeurIPS 2025 poster_

### Official Review · Reviewer_Ueg5 · 2025-06-26

**Clarity:** 3
**Significance:** 4
**Originality:** 3
**Rating:** 5
**Confidence:** 3

**Summary:**

This paper investigates an entropic-regularized optimal transport (OT) problem between a continuous source distribution and a discrete target distribution--referred to as the **semi-OT** problem. Specifically, the authors focus on the dual formulation of this problem, which reduces to a stochastic optimization task in a finite-dimensional space due to the discreteness of the target distribution.

To solve this, the authors propose an algorithm named **DRAG**, designed with the following components:

1. The core of the algorithm is stochastic gradient descent (SGD).
2. The entropic regularization parameter $\gamma$ is gradually decreased as more samples are observed in an online fashion.
3. The learning rate is annealed over time -- together with the regularization parameter.
4. A projection step is applied right after each gradient update.

The authors prove that, with appropriate annealing for the regularization parameter and step size, **DRAG** achieves optimal convergence rates:
- $\mathcal{O}(1/t)$ for both the OT potential and OT cost,
- $\mathcal{O}(1/\sqrt{t})$ for the OT map.

These convergence rates are also empirically verified in synthetic numerical experiments. Furthermore, the experiments show that DRAG is able to find accurate OT maps in practice.

**Questions:**

Some questions/suggestions:

- In section 2, the dependence of OT quantities on the cost function $c$ should be dropped if the cost is already quadratic (for example, right after line 102, right after right 107). Otherwise, these equations should be in general forms with $c$ in their formulations.

- The projection step looks unconventional to me. Although it does not hurt to project back to the set that is known to contain the optimal solution, is necessary to do so in terms of theory?

- The paragraph from line 200 to line 202 will benefit from further details and discussion.

**Ethical Concerns:**

["NO or VERY MINOR ethics concerns only"]

**Final Justification:**

The authors have sufficiently addressed my comments. I therefore keep my score.

**Limitations:**

yes

**Quality:**

3

**Strengths And Weaknesses:**

# Strengths

- The theoretical findings are solid: while previous methods with a fixed regularization parameter $\epsilon$ has a complexity dependent adversely on $\epsilon^{-1}$, i.e., $O(\epsilon^{c} t^{-b})$, DRAG -- with a careful design of the sequence of decreasing regularization parameters -- bypasses this dependence.

- The algorithmic design of DRAG is desirable in the context of large-scale learning. Not only it unbiasedly converges to the optimal solution (OT map / OT potential), it is also doing so in an online manner -- using one sample or a minibatch of samples at each iteration.

# Weaknesses

- Assumption 1 requires the target distribution to be discrete with equal weights, which limits the applicability of DRAG. In the experiment on synthetic data, the target distribution in Example 2 has unequal weights. It is unclear to me how DRAG is still valid for this case.

- In the theoretical results, to obtain the optimal convergence rate, they require $a$ to be (very) close to $1/3$ and $b=2/3$ to ensure the flexibility of $\alpha$. However, this choice has a negative impact. For example, in the inequality right after line 220, the power of $t$ is at the limit of $0$, making that claim vacuous. My impression is that this choice has an overall (negative) trade-off in the optimization bounds and the complexity (that is probably ignored by the asymptotic notations) that deserves a careful discussion.

- I would expect more data sets to be considered in the Generative modeling task and the Monge-Kantorovich Quantiles experiment. Currently, each experiment has only two datasets, which makes the conclusion on the performance of DRAG premature.

---

> ### Author Rebuttal · Authors · 2025-07-29
>
> Dear Reviewer,
> Thank you for your positive and constructive feedback on our work. We are particularly pleased that you highlighted the convergence guarantees for decreasing regularization in OT and the use of online algorithm that is suitable for large-scale problems.
> We address the specific concerns below:
>
> ### **Weaknesses**
>
> 1. We thank the reviewer for this insightful question.
> It is correct that our theoretical analysis assumes uniform weights for the target distribution, as stated in Assumption 1. This restriction was made primarily to simplify the convergence proof. However, we believe that the results should still hold when the target distribution has strictly positive but unequal weights, although extending the theory to this more general setting would require additional technical work.
> In the revised version, we will make this point clearer and explicitly state that Example 2 was included to illustrate that DRAG continues to perform well in practice even when the uniformity assumption is relaxed.
>
> 2.  Since the result of Proposition 2 holds for any $q$, there is no negative trade-off: we can always make the decay arbitrarily fast by increasing $q$, even when $2a$ is close to $b$. Indeed, given any $\delta > 0$ and any constants $a, b$ such that $2a < b$, and for any $s > 0$, we can always choose $q$ large enough so that
> $$
> -q(b - 2a) + \delta < -s.
> $$
> We will clarify this point in the revised version and emphasize that there is no trade-off in the choice of $a$ and $b$ here, thanks to the fact that the result holds for all $q > 0$.
>
> 3. We agree that the number of datasets considered in both the generative modeling and Monge-Kantorovich quantiles experiments is currently limited, and that more examples would help strengthen our conclusions.
> To address this, we have already extended both experimental sections and will include the results in the appendix of the revised version. In particular, we now provide more comprehensive examples involving 16 different measures for both the generative modeling and MK quantile tasks. Moreover, we use discrete measures with up to $10^8$ points for MK quantiles, emphasizing that DRAG, as an SGD-based algorithm, is well suited for large-scale problems.
> While the rebuttal process does not allow us to include additional figures, we summarize the findings here:
> - For the generative modeling task, DRAG consistently outperformed Adam-based optimization, especially on more complex synthetic measures, where standard SGD showed instability.
> - For the MK quantile task, the general observations remained the same: Neural OT performed significantly worse than DRAG , both in terms of coverage and bias, despite being significantly slower. DRAG continued to outperform Online Sinkhorn in terms of accurately covering the support of the target distribution without entropic bias, which aligns with the theoretical expectation of convex MK regions.
> We will make sure to discuss these additional results in the final version to provide a more complete empirical validation.
>
>
>
>
> ### **Questions**
>
> 1. We thank the reviewer for the suggestion. In the revised version, we will present results in general form with explicit cost dependence when possible, especially when introducing the background on (entropic) Optimal Transport in Section 2. We will then specialize to the quadratic cost case only when needed, to enhance clarity and avoid unnecessary notational overload.
>
> 2. The projection step in Algorithm 1 is indeed a thresholding operation. This step is crucial in the convergence analysis of the non-averaged iterates of DRAG, as established in Theorem 1. In practice, we also find this projection step helpful, particularly during the early iterations, where it can improve both the convergence speed and the stability of the algorithm.
>
> 3.  The notational framework on lines 200–203 was intended to emphasize that the optimal potential $\mathbf{g}^{\*}$ is only unique up to transformations of the form $\mathbf{g}^{\*} + a \mathbf{1}_M$, with $a \in \mathbb{R}$. Therefore, we conduct all our convergence analysis in the orthogonal complement of $\mathrm{vect}(\mathbf{1}_M)$. We apologize for the lack of clarity and will provide a clearer explanation in the final version of the paper.

---

> > ### Comment · Reviewer_Ueg5 · 2025-08-04
> >
> > Thanks for addressing my comments. I maintain my score.

---

### Official Review · Reviewer_zxMk · 2025-06-30

**Clarity:** 3
**Significance:** 2
**Originality:** 2
**Rating:** 4
**Confidence:** 3

**Summary:**

This paper introduces DRAG, a stochastic gradient algorithm for solving semi-discrete optimal transport problems.  DRAG employs a decreasing entropic regularization schedule that shrinks with the number of optimization steps and achieves:

1. $O(1/t$) convergence for OT cost and potentials,

2. $O(1/\sqrt{t})$ convergence for the transport map,

The convergence is  independent on the vanishing regularization parameter.

**Questions:**

1. Is the averaging step for the gradient  $
\bar g_t = \frac{1}{1+t} g_t + \frac{t}{t+1} \bar{g}_{t-1} $
essential for convergence, stability, or theoretical guarantees?  How it impacts empirical performance?

 2.We noticed the following potential typos and would like to confirm:

line 111: $T_{\mu,\nu}(x)=...=x-y_j$ should be  $T_{\mu,\nu}(x)=...=y_j$,

line 241: $\alpha \in (0,1]$ should be $\alpha \in (\frac{1}{2},1]$.

**Ethical Concerns:**

["NO or VERY MINOR ethics concerns only"]

**Final Justification:**

**Update**
The authors' response addressed my primary concern, I have updated my score accordingly.

**Limitations:**

yes

**Paper Formatting Concerns:**

null

**Quality:**

2

**Strengths And Weaknesses:**

Strengthens:

1. The DRAG algorithm is conceptually simple and easy to implement.

2.The restricted strong convexity argument is technically interesting.

3.Relevant experiments to support claims are included.

Weakness:

1.Some related ideas (e.g., multiscale Sinkhorn, continuation methods) have been considered in fully discrete OT; a clearer positioning relative to those methods would help sharpen the originality claim.

2.Scalability of the method is not rigorously evaluated beyond low dimension ( dimension 10 is shown, but real-world applications could go higher).

---

> ### Author Rebuttal · Authors · 2025-07-29
>
> Dear Reviewer,
> We would like to thank you for your time and thoughtful comments. Please find below our responses to the weaknesses and questions you raised.
>
>
> ### **Weaknesses**
>
> 1.  **Originality of DRAG**
>
> With our algorithm DRAG, and in the semi-discrete setting, we provide, to the best of our knowledge, the first theoretical proof of accelerated convergence rates due to decreasing regularization for solving an OT problem in any setting. While the heuristic of $\varepsilon$-scaling / $\varepsilon$-annealing [1] in discrete OT has been observed in practice to enhance the convergence rate of the Sinkhorn algorithm, theoretical guarantees that a decreasing regularization scheme can indeed accelerate the convergence of an OT algorithm remained an open question prior to our work.
>
> Whereas algorithms with fixed regularization typically suffer from adverse dependence on the regularization parameter, our use of a decreasing regularization sequence entirely avoids this issue.
> Moreover, unlike discrete solvers that reuse all samples multiple times (multi-pass), DRAG is adaptive to the number of samples and operates in an online (one-pass) setting, without the need to store past samples. This enables us to solve the semi-discrete OT problem without incurring either discretization or regularization bias.
>
> In the final version, we will further emphasize these distinctions, highlighting DRAG’s main contribution: it is the first OT algorithm to confirm, through theory, the heuristic (previously only observed in practice) that decreasing regularization leads to acceleration.
>
>
> 2. **Scalability of the method** (This reply is mostly the same for both Reviewer fbxE and Reviewer zxMk.)
>
> We reran our synthetic data experiments (Examples 1, 2, and 3) in dimensions 100 and 1000, instead of 10. We observed the same convergence rate behavior across all experiments with respect to the number of samples, with similar final errors. The experiments were approximately 4 and 15 times slower to run, respectively, which is solely due to the increased time required to compute $c(x, y) = \frac{1}{2}||x - y||^2$ on our GPU.
> This similar behavior in higher dimensions is not surprising in the semi-discrete OT setting, which is known not to suffer from the curse of dimensionality, as shown in [2]. While changing the parameter $d$ does not affect the error in our setting, increasing the number of points $M$ does impact the convergence rate, as we are solving a stochastic optimization problem in $\mathbb{R}^M$ via the semi-dual formulation.
> Since our algorithm is SGD-based and has linear memory complexity $\mathcal{O}(M)$ and computational complexity $\mathcal{O}(dM)$, we believe it is highly scalable.
> We will include these comments in the final version and thank the reviewer for highlighting this point. Furthermore, we will replace our synthetic examples in Figures 1, 2, and 3 with their dimension-1000 versions to better illustrate this scalability.
>
> ### **Questions**
>
> 1. The averaging of the gradient steps is indeed essential, both theoretically and numerically.
> **Theoretically:** Averaging the iterates leads to improved convergence guarantees, as shown in Theorem 2. For instance, for any $\alpha > 1/2$, choosing parameters $a = \frac{1}{3}^-$ and $b = 2/3$ results in a convergence rate of only $\mathcal{O}(1/t^b)$ for the non-averaged iterates, compared to the optimal rate of $\mathcal{O}(1/t)$ for the averaged iterates. Moreover, our convergence analysis requires $b < 1$, which prevents us from achieving the optimal $\mathcal{O}(1/t)$ rate for non-averaged iterates in theory.  Note that acceleration by averagin is classic in the stochastic gradient descent litterature [2].
> **Numerically:** This difference is also observed in practice. The non-averaged iterates converge at a rate of approximately $\mathcal{O}(1/t^b)$, while the averaged iterates converge faster and exhibit improved stability. To illustrate this, we will include the errors of the non-averaged iterates in Figure 1, showing that averaging indeed enhances convergence in practice.
>
> 2. We confirm that these are indeed typos, and we thank you for pointing them out.
>
>
> We hope that we have addressed all of your concerns and that the clarifications and improvements above provide a clearer picture of the quality, originality, and significance of our work.
>
> ### **References**
>
> [1] B. Schmitzer. Stabilized sparse scaling algorithms for entropy regularized transport problems. SIAM Journal on Scientific Computing
>
> [2] Polyak, B. T., & Juditsky, A. B. (1992). Acceleration of stochastic approximation by averaging. SIAM Journal on Control and Optimization, 30(4), 838–855.

---

> > ### Comment · Reviewer_zxMk · 2025-08-04
> >
> > Thank you for the detailed comments and added experiments. This cleared up a lot of my questions, and I've decided to raise my score 3->4 as I think the paper has an interesting contribution.

---

### Official Review · Reviewer_XDRF · 2025-07-01

**Clarity:** 3
**Significance:** 4
**Originality:** 3
**Rating:** 5
**Confidence:** 3

**Summary:**

The authors study the unregularized optimal transport (OT) problem with quadratic cost in the semi-discrete setting, where the source measure is continuous and the target measure is discrete. They focus on the semi-dual formulation, which yields a convex optimization problem, and analyze the convergence properties of single-pass (online) stochastic gradient descent algorithms applied to the entropy-regularized formulation when the regularization parameter is reduced during the iterations according to an annealing schedule. When averaging over iterations, the authors establish convergence rates of O(1/t) for both the OT cost and potential, and O(1/\sqrt{t}) for the OT map, where t denotes the iteration count. These results hinge on proving that a regularization-free restricted strong convexity property holds locally within a ball around the optimum, and on showing that the algorithm's iterates remain within this region with high probability. Numerical experiments on synthetic data are provided to support the theoretical findings.

**Questions:**

A) In the related works section of the introduction, the authors write that “decreasing the regularization is not only computationally beneficial but also statistically necessary,” referring to the results in [45]. However, as far as I can tell, [45] does not show that decreasing regularization is truly necessary. They only show that a method using this approach can reach the best possible rates, which is just one way to succeed, not the only one. Could the authors please clarify what kind of necessity they had in mind?

B) Proposition 2 (before averaging) seems to require some restrictions on the Hölder continuity parameter \alpha, e.g. 1/2 < \alpha \leq 1, while Theorem 2 (which uses averaging) seems to allow for weaker conditions, e.g. 0 < \alpha \leq 1. Could the authors explain why averaging leads to this improvement?

C) Overall, I feel the paper does not give enough intuition about why the restricted strong convexity property can hold locally with a parameter that does not depend on the regularization size. Since this seems to be one of the main ideas of the work, I think it deserves a clearer explanation. Could the authors say more about this?

**Ethical Concerns:**

["NO or VERY MINOR ethics concerns only"]

**Final Justification:**

I appreciate the authors’ rebuttal, which addressed my questions. I believe this is a good paper and have maintained my positive score.

**Limitations:**

Yes.

**Paper Formatting Concerns:**

Bi,

**Quality:**

4

**Strengths And Weaknesses:**

Quality and Clarity: The paper is clear and easy to follow, with the main results introduced and explained well. The key ideas are presented in a direct way, and the numerical experiments are easy to interpret. It would be helpful to include more explanation of what causes the main effect being studied (on restricted strong convexity).

Significance: This work seems to provide the first (?) theoretical guarantees for annealing stochastic iterative schemes in optimal transport.

Originality: While the paper seems to bring together the idea of decreasing regularization (as in [45]) with a stronger analysis of SGD building from [5], many of the core ideas, specifically showing that a local restricted strong convexity property holds with regularization-independent parameters—seems both original and non-trivial.

---

> ### Author Rebuttal · Authors · 2025-07-29
>
> Dear Reviewer,
> Thank you for your positive and constructive feedback on our work. We are particularly pleased that you highlighted the convergence guarantees for decreasing regularization in OT. We also believe this is the first such result. We appreciate your insightful questions and helpful suggestions.
>
> ### **Questions**
> A) We apologize for the lack of clarity. What we intended to convey is that, when using entropic OT as in [45], decreasing the regularization with respect to the number of points is necessary to achieve the optimal convergence rate. Using a larger, fixed regularization would lead to a non-optimal bias.
> Of course, an estimator that does not rely on entropic OT could, in principle, also achieve the optimal rate. Although, to the best of our knowledge, no such estimator currently exists for the semi-discrete OT setting. Therefore, while decreasing regularization is necessary when using entropic OT, using entropic OT itself may not be strictly necessary. We will better explain our meaning in the revised version of the paper.
>
> B) We believe this question may have arisen due to a typo in the comment following the statement of Theorem 2, on line 241, where it incorrectly states $\alpha \in (0,1]$ instead of the correct $\alpha \in (1/2, 1]$. This typo was corrected in the full version of the paper included in the supplementary material. We apologize for the inconvenience.
>  Both theorems allow for $\alpha \in (0,1]$ and yield a convergence rate. There is indeed an improvement in the convergence rate for the averaged case: when $\alpha > 1/2$, choosing $a = 1/3^-$ and $b = 2/3$ leads to a convergence rate of $\mathcal{O}(1/t^{2/3})$ for the non-averaged iterates, compared to the optimal rate of $\mathcal{O}(1/t)$ for the averaged iterates.
> This acceleration due to averaging is well known in the context of SGD algorithms [1], where, without averaging, the iterates tend to oscillate around the minimizer due to stochastic noise, while averaging “smooths out” this noise by aggregating information from multiple iterates.
>
> C) We acknowledge that restricted strong convexity is a key property for the success of the convergence analysis of DRAG. In our setting, it holds locally with a parameter that does not depend on $\varepsilon$, because, denoting by $\rho^\varepsilon_*$ the smallest eigenvalue of the Hessian of $H_\varepsilon$ on $\mathrm{vect}(\mathbf{1})^\bot$, we have a uniform lower bound $\rho_* = \inf_{\varepsilon \in (0,1]} \rho^\varepsilon_*$. In particular, we do not have $\rho^\varepsilon_* \to 0$ as $\varepsilon \to 0$.
> This is due to the fact that, under our assumptions, even the non-regularized semi-dual has $\rho^0_* > 0$. This provides a form of local strong convexity, stemming from the regularity of the quadratic cost and the uniqueness of the minimizer. Using Theorem 3.2 in [2], we have $\rho^\varepsilon_* \geq \frac{1}{2}\rho^{\*}$ for all $\varepsilon \in [0,1]$.
> However, while the local restricted convexity is common for all $\varepsilon$, we see that the area in which it is true depends on $\varepsilon$, which thus show that when we decrease the regularization, the radius around $\mathbf{g}_{\varepsilon}^*$ in which we have a favorible restrcited strong convexity is smaller.
> We hope this explanation clarifies the local behavior, and we will incorporate it into the final version of the paper.
>
> ### **References**
> [1] Polyak, B. T., \& Juditsky, A. B. (1992). Acceleration of stochastic approximation by averaging. SIAM Journal on Control and Optimization, 30(4), 838–855.

---

> > ### Comment · Reviewer_XDRF · 2025-08-02
> >
> > I appreciate the authors’ rebuttal, which addressed my questions. Having reviewed the other evaluations and responses, I will maintain my score for now and await further discussion with the other reviewers.

---

### Official Review · Reviewer_fbxE · 2025-07-02

**Clarity:** 3
**Significance:** 2
**Originality:** 2
**Rating:** 4
**Confidence:** 2

**Summary:**

In this paper, the authors propose a new algorithm for solving semi-discrete optimal transport problem, which reduces the regularization parameter while increasing the number of samples from the source measure. The authors provide and prove the estimates for convergence rate of potentials and optimal mappings. The authors provide results of working their algorithms and compare with Online Sinkhorn [1] and Neural OT [2].

**Questions:**

- Typos?
    - (l.153) It is not necessary to assume that a continuous function is bounded on a compact set, isn't it?
    - To what extent does the notational framework presented between lines 202-203 enhance the clarity of the mathematical exposition?
- Does your analysis of the decreasing regularization scheme generalize to the setting of continuous target measures? Does your scheme of decreasing epsilon while increasing the number of training samples is viable in continuous target measure case?
- Can the α-Hölder continuity assumption on the density be relaxed while maintaining the convergence properties? Furthermore, does there exist a counterexample demonstrating that this regularity condition cannot be entirely removed?
- For which costs do the results of the paper remain valid?

**Ethical Concerns:**

["NO or VERY MINOR ethics concerns only"]

**Final Justification:**

*Why not higher score?* DRAG has certain restrictions limiting its practical usability. 1) DRAG is designed only for the quadratic cost, 2) it seems to be hard to generalize it for the case of continuous distributions, 3) its scalability remain questionable for me.

*Why not lowet score?* The paper proposes a new approach and presents a good combination of its empirical verification and supporting theoretical results.

**Limitations:**

The authors have addressed some limitations of their approach in the conclusion. But the authors don’t specify for which costs the results of the paper remain valid. This aspect should be stated as a limitation of their approach.

**Paper Formatting Concerns:**

No major formatting concerns.

**Quality:**

2

**Strengths And Weaknesses:**

**Strengths.**

The paper is almost clearly written and provides a complete theoretical analysis of the convergence rate of optimal mappings and potentials.

**Weaknesses.**

My first concern is devoted to the quality and clarity of the theoretical results. The proofs of many lemmas given in the paper are quite technical, they reduce to consideration of many particular cases, which complicates their verification, and does not allow me to say with full confidence that all the proofs given are completely correct and original. Besides, I am confused by the fact that all of the results only cover the case of the quadratic cost. The line 84 states that some of the results can be extended to other costs, but authors do not specify which exactly and do not state it as a limitation of their approach. I kindly suggest the authors fix this point.

Besides, I am concerned by the limited experimental validation of the proposed algorithm. All of the experiments are conducted in small-dimensional (with dimensionality not exceeding ten) spaces and cover only toy artificial setups, see their Section 5. Thus, It is not obvious for me whether the presented algorithm is scalable or not. I kindly suggest the authors to include some experiments testing their approach  in bigger dimensions to mitigate these concerns. Besides, I am wondering what real-world problems such an algorithm could be used for? I would appreciate it if the authors could conduct or list the real-world experiments where their approach is really effective in comparison with other well-known algorithms.

Additionally, although this paper is devoted to semi-discrete optimal transport, I am wondering whether the proposed scheme of decreasing epsilon while increasing the number of training samples is viable in the continuous target measure case. The authors have not mentioned this, but I would like to know whether semi-discreteness is a fundamental limitation of this method or not.

**To conclude,** the overall contribution of the paper is questionable for me. Theoretical estimation of the convergence rate (Theorem 3) for optimal mappings is asymptotically equivalent to the estimation from [3] (where it holds for all $\alpha\in(0,1)$ while the results of this paper hold for $\alpha\geq 1/2$). Thus, I have doubts regarding the originality of the theoretical results presented in the paper as long as practicality and real-world applications of the developed approach.


*Typos:*

- The formulation of the dual problem (2) contains typos affecting the correctness of the formulation. Supremum must be used in formula (2) instead of minimum by continuous functions, because it is claimed that expression (1) is equal to expression.

**References.**

[1] A. Mensch and G. Peyré. Online sinkhorn: optimal transport distances from sample streams. In Proceedings of the 34th International Conference on Neural Information Processing Systems, 2020.

[2] A. Korotin, D. Selikhanovych, and E. Burnaev. Neural optimal transport. The 11th International Conference on Learning Representations, 2023.

[3] A.-A. Pooladian, V. Divol, and J. Niles-Weed. Minimax estimation of discontinuous optimal transport maps: The semi-discrete case. In the International Conference on Machine Learning, 2023.

---

> ### Author Rebuttal · Authors · 2025-07-29
>
> Dear Reviewer,
> Thank you for your detailed feedback and constructive comments. We also thank you for pointing out some typos such as the one regarding equation (2), they will be corrected in the revised version.
> We adress each point below.
>
> ### **Weaknesses**
>
> 1.   We acknowledge that several of the proofs are technically involved, and we sincerely appreciate the time and attention you dedicated to reading through them. We would like to emphasize that, to the best of our knowledge, the proofs are both correct and original, and we have carefully verified their correctness. We appreciate the reviewer's concern and would welcome any specific suggestions that could help improve the accessibility of our proofs.
>
> 2. Our results indeed cover only the quadratic cost case, as we rely on Corollary 2 in [1] concerning the convergence of entropic potentials, which we stated in Proposition 1. This result applies only to the quadratic cost. Note that [2] also focuses exclusively on the quadratic cost, as their analysis is similarly based on [1].
> That said, the two key technical lemmas: Lemma 1 (projection step) and Lemma 2 (Restricted Strong Convexity) can be generalized to other cost functions. In particular, the lower bound on $\rho_*$ stated on line 213 is proven for a broader class of costs in Proposition 5.1 of [3].
> Proposition 1, which is restricted to the quadratic cost, is therefore the limiting factor. Generalizing it to other costs would directly enable us to extend our results without modifying the convergence algorithm or the structure of our proof. However, we believe this generalization is technically challenging and constitutes an interesting direction for future work.
> In the final version, we will further emphasize that Proposition 1 is the limiting factor and will explicitly state the generalization to other costs as a valuable research direction.
>
> 3.  (This reply is mostly the same for both Reviewer fbxE and Reviewer zxMk) We reran our synthetic data experiments (Examples 1, 2, and 3) in dimensions 100 and 1000, instead of 10. We observed the same convergence rate behavior across all experiments with respect to the number of samples, with similar final errors. The experiments were approximately 4 and 15 times slower to run, respectively, which is solely due to the increased time required to compute $c(x, y) = \frac{1}{2}||x - y||^2$ on our GPU.
> This similar behavior in higher dimensions is not surprising in the semi-discrete OT setting, which is known not to suffer from the curse of dimensionality, as shown in [2]. While changing the parameter $d$ does not affect the error in our setting, increasing the number of points $M$ does impact the convergence rate, as we are solving a stochastic optimization problem in $\mathbb{R}^M$ via the semi-dual formulation.
> Since our algorithm is SGD-based and has linear memory complexity $\mathcal{O}(M)$ and computational complexity $\mathcal{O}(dM)$ per iteration, we believe it is highly scalable.
> We will include these comments in the final version and thank the reviewer for highlighting this point. Furthermore, we will replace our synthetic examples in Figures 1, 2, and 3 with their dimension-1000 versions to better illustrate this scalability.
>
> 4. For real-world applications, DRAG can be used in generative modeling tasks that rely on semi-discrete OT to improve performance, as in [4] with GANs, [5] with diffusion models, and in statistics for Monge–Kantorovich quantile estimation [7].
> Besides the fact that DRAG has optimal convergence guarantees for semi-discrete OT, we also observed in practice its better performance compared to Adam, which is not tailored for the OT task. As mentioned in our reply to Reviewer Ueg5’s request, we have extended both the generative modeling and Monge–Kantorovich quantile experiments to include more complex and large-scale examples.
> Furthermore, unlike DRAG, Adam does not come with theoretical guarantees for convergence to the true OT potential and map, to the best of our knowledge.
>
> ### **Questions**
>
> 1. a) (l. 153) You are correct, the assumption is unnecessary.
> b) The notational framework on lines 202–203 was intended to emphasize that the optimal potential $\mathbf{g}^*$ is only unique up to transformations of the form $\mathbf{g}^{\*} + a \mathbf{1}_{M}$, with $a \in \mathbb{R}$. Therefore, we conduct all our convergence analysis in the orthogonal complement of $\mathrm{vect}(\mathbf{1}_M)$.
>
> 2. While extending our results to the continuous setting is an interesting direction for future work, our method fundamentally relies on the fact that the semi-discrete setting leads to a finite-dimensional problem. This property does not hold in the fully continuous setting. Therefore, the semi-discrete nature of the problem is indeed a limitation of our approach.
> That said, we would like to emphasize that the semi-discrete setting is, in itself, a very interesting and challenging problem with numerous applications. For example, it arises in generative modeling [4,5] and statistics, such as with Monge–Kantorovich quantiles.
>
> 3. While our algorithm achieves the best convergence rate when $\mu$ is $\alpha$-Hölder with $\alpha > 1/2$, we still guarantee convergence for any $\alpha$, as stated in Theorem 2. The condition $\alpha > 1/2$ is somewhat conservative and arises from our convergence analysis.
> That said, we believe that at least continuity of the source measure is necessary, since without that the optimal potential $\mathbf{g}^{\*}$ may not be unique, even in the orthogonal complement of $\mathrm{vect}(\mathbf{1}\_{M})$. For example, consider the case where $\mu = \frac{1}{2}\mathcal{U}([-2, -1]) + \frac{1}{2}\mathcal{U}([1, 2])$ and $\nu = \frac{1}{2}\delta_{-1} + \frac{1}{2}\delta_{1}$. Then, for the quadratic cost, all vectors of the form $\mathbf{g}_{x} = (-x, x)$ with $x \in (-1, 1)$ are optimal for the semi-dual problem, even though they all lie in the orthogonal of $\mathrm{vect}(\mathbf{1}_M)$.
>
> 4. (Cost question) see weakness 2.
>
> ### **Comparison with [2]**
>
>  While [2] provides statistical guarantees for the approximation of the semi-discrete OT map and establishes that an optimal rate of $\mathcal{O}(1/\sqrt{t})$ is achievable, this is done by discretizing the source measure using $t$ points and employing the discrete entropic OT map as an estimator, with entropic regularization $\varepsilon \simeq 1/\sqrt{t}$.
>
> Their work offers strong statistical guarantees for OT map estimation, provided that the discrete OT map is accurately computed. However, it focuses on an offline setting where all $t$ samples from the source measure are stored, and a discrete entropic OT problem is solved. The analysis is limited to sample complexity, specifically the statistical bias between the semi-discrete and discrete entropic OT maps. The computational complexity of Sinkhorn required to accurately approximate the OT map is not discussed, even though it typically scales poorly with $\varepsilon^{-1}$. Moreover, in this offline or multi-pass setting, increasing the number of Sinkhorn iterations does not correspond to using more samples.
>
> In contrast, our paper directly tackles the semi-discrete OT problem without introducing discretization bias. We propose an algorithm that operates in an online (single-pass) setting, does not require storing past samples from the source measure, and demonstrates that decreasing regularization can indeed accelerate convergence for OT algorithms. Their paper provides stronger statistical guarantees, achieving optimal rates for $\alpha \in (0,1)$, whereas our results apply to $\alpha > 1/2$. However, we directly solve the semi-discrete problem, avoid discretization bias, and offer a scalable algorithm with linear computational and memory complexity per iteration.
>
> ### **Overall contribution of the paper**
> We would like to emphasize that, while decreasing regularization is a popular heuristic in optimal transport, DRAG is, to the best of our knowledge, the first algorithm in OT where acceleration due to decreasing regularization is formally proven. Prior to our work, theoretical guarantees that a decreasing regularization scheme can indeed accelerate the convergence of an OT algorithm remained an open question.
> Whereas a fixed regularization typically leads to a complexity that depends adversely on $\varepsilon^{-1}$ in OT algorithms, our approach achieves convergence rates without such adverse dependence, thanks to the careful design of our regularization schedule. This theoretical benefit is reflected in practice, as shown in Figure 2, where DRAG outperforms all fixed regularization baselines.
>
> We hope that our response has clarified the originality and significance of our work. We appreciate the reviewer’s time and hope they find the revised explanations helpful.
>
> ### **References**
>
> [1] Delalande, A. (2022). Nearly tight convergence bounds for semi-discrete entropic optimal transport.  International Conference on Artificial Intelligence and Statistics (pp. 1619-1642). PMLR
>
> [2] A.-A. Pooladian, V. Divol, and J. Niles-Weed. Minimax estimation of discontinuous optimal transport maps: The semi-discrete case. In the International Conference on Machine Learning, 2023.
>
> [3] Chizat, L., Delalande, A., \& Vaškevičius, T. (2024). Sharper Exponential Convergence Rates for Sinkhorn's Algorithm in Continuous Settings. arXiv preprint arXiv:2407.01202.
>
> [4] An, D., Guo, Y., Lei, N., Luo, Z., Yau, S. T., \& Gu, X. (2019). AE-OT: A new generative model based on extended semi-discrete optimal transport. ICLR 2020.
>
> [5] Li, Zezeng, et al. "DPM-OT: a new diffusion probabilistic model based on optimal transport." Proceedings of the ieee/cvf international conference on computer vision. 2023.

---

> > ### Comment · Area_Chair_nu8Z · 2025-08-05
> > **Rebuttal response required**
> >
> > Dear reviewer,
> > Please engage with the authors rebuttal.
> > Thanks,
> > Your AC

---

> > ### Comment · Reviewer_fbxE · 2025-08-05
> > **Thank you for the answers**
> >
> > I thank the authors for their clarifications and additional experiments. However,  I remain confused by the scalability  of the proposed approach, since the new experiment was conducted in 1000-dimensional space which seems to be not that big. This immediately raises question regarding the overall significance of the proposed approach since it has certain restrictions limiting its practical usability. That is, 1) DRAG is designed only for the quadratic cost, 2) it seems to be hard to generalize it for the case of continuous distributions, 3) its scalability remain questionable for me.
> >
> > Still, since the authors provide necessary clarifications on my questions and conduct additional related experiments, I increase my score to 4.

---

> ### Author Response · Authors · 2025-08-08
> **Thank you for your reply**
>
> Dear Reviewer,
> Thank you for your follow-up and for increasing your score based on our clarifications and additional experiments. We truly appreciate your engagement with our work.
>
> We understand your concerns about scalability. While 1000 dimensions may not appear large by machine learning standards, it corresponds to the true intrinsic dimension of the data in our setting. Our goal was to demonstrate the absence of the curse of dimensionality. If the curse were present, we would expect a convergence rate of the form $\mathcal{O}(n^{-1/d})$, which deteriorates with increasing dimension. Instead, we observe a convergence rate of $\mathcal{O}(n^{-1/2})$, which is independent of the dimension and thus highlights that semi-discrete OT avoids this curse. We acknowledge that optimal transport can be applied in high-dimensional ambient spaces; however, in practice, the data often lies on a low-dimensional manifold, which enables favorable convergence behavior [1].
>
> You are right that our theoretical results currently apply only to the quadratic cost, corresponding to the 2-Wasserstein distance. While this is indeed a restriction, we note that the quadratic cost plays a central role in optimal transport theory and has deep connections to PDEs, geometry, and many practical applications. Although DRAG can be used with other cost functions without modification, extending the theoretical analysis beyond the quadratic case is an interesting direction for future work.
>
> As for the continuous setting, we agree that our focus is on semi-discrete OT. Still, the ability of DRAG to operate online, with only one distribution discretized and without storing past samples, contrary to discrete solvers, makes it a practical proxy for large-scale continuous problems, especially given its linear memory usage in the number of discretized points.
>
>
> **Reference:**
> [1] Weed, J., & Bach, F. (2019). Sharp asymptotic and finite-sample rates of convergence of empirical measures in Wasserstein distance. Bernoulli, 25(4A), 2620-2648.

---

### Official Review · Reviewer_nRvS · 2025-07-02

**Clarity:** 3
**Significance:** 4
**Originality:** 4
**Rating:** 5
**Confidence:** 3

**Summary:**

Entropy regularisation is one of the most popular methods for computing optimal transport distances. Adding entropy decreases computational time however worsens the approximation. Scaling the amount of entropy to zero should therefore reduce computational time whilst providing a better estimate for the OT distance. This paper analysis the affect of entropy and proposes a scaling of the entropic regularisation which produces a consistent estimate of the OT distance with a rate of convergence.

Disclaimer: due to the length of the paper and the time available to review all NeurIPS submissions I did not check the proofs.

**Questions:**

See 1-3 in the strengths/weaknesses. And
4. In Proposition 1, does the constant $K_0$ depend *only* on $\nu$?

**Ethical Concerns:**

["NO or VERY MINOR ethics concerns only"]

**Final Justification:**

My comments and scores seem to be inline with the other reviewers so I'm comfortable with my assessment. In short, I thought the results were good and interesting. This is a problem where I'm only aware of heuristics and a more principled approach is valuable. I recommend acceptance.

**Limitations:**

Yes

**Paper Formatting Concerns:**

None.

**Quality:**

4

**Strengths And Weaknesses:**

The theory (assuming it is correct) is very nice. I haven't seen this type of result before (although I only work adjacent to the field) and it is definitely useful. The only scaling results I am aware of are heuristic results. The numerical results are good, but the main contribution (in my opinion) is theoretical.

There were a few parts where I felt the paper was harder to understand than it should have been. For example:
1. In eq (1) $\frac{d\pi}{d\mu d\nu}$ is difficult for me to understand. I assume it means $\frac{d\pi}{d \mu\otimes \nu}$.
2. Was the scaling of $\gamma_k$ in Algorithm 1 ever explained?
3. How important is the projection in Algorithm 1? I'm understanding it as a thresholding step if $g_{k-1} - \gamma_k \nabla_g h_{\eps_{k-1}}(X_k,g_{k-1}))$.

---

> ### Author Rebuttal · Authors · 2025-07-29
>
> Dear Reviewer,
> Thank you for your positive evaluation of our work and for your questions. We are also glad that you appreciate that our work is, to the best of our knowledge, the first to provide theoretical guarantees for acceleration through a decreasing regularization scheme in OT, whereas this approach had previously been used in the discrete setting only as a heuristic.
>
> Please find below our replies to the weaknesses you pointed out and the questions you raised.
>
> ### **Weaknesses**
>
> 1. You are right, it should be $\frac{d\pi}{d\mu \otimes d\nu}$. We will incorporate this correction in the final version.
>
> 2. While $\gamma_k = \frac{\gamma_1}{k^b}$ with $\gamma_1, b > 0$ is the standard decreasing step size (learning rate) sequence in the SGD algorithm, as in [1, Equation (1)], it is true that in our case, the decay parameter $b$ depends on $a$ and $\alpha$ to achieve the best convergence rate. This follows from the convergence analysis of DRAG in Theorem 2, where the optimal rate is always attained for $b = \frac{2}{3}$ as soon as $\alpha > 1/2$, as discussed in the paragraph "Dependence on $a$, $b$, and $\alpha$" (l. 224).
> In the final version, we will emphasize that the specific scaling of $\gamma_k$ is motivated by the convergence proof of DRAG.
>
> 3. The projection step in Algorithm 1 is indeed a thresholding operation. This step is crucial in the convergence analysis of the non-averaged iterates of DRAG, as established in Theorem 1. In practice, we also find this projection step helpful, particularly during the early iterations, where it can improve both the convergence speed and the stability of the algorithm.
> In the later iterations, when the estimator $\mathbf{g}_t$ is close to $\mathbf{g}^*$, the thresholding becomes unnecessary and typically does not occur. In the final version, we will further emphasize the usefulness of this projection step, both from a theoretical perspective and in practice, especially in the early stages of the algorithm.
>
> ### **Question on Proposition 1**
> $K_0$ depends not only on the target measure $\nu$, but also on the source measure $\mu$, through the radius $R$ and the constants $m_1, m_2 > 0$ such that $m_1 < d\mu < m_2$. We will update Proposition 1 to reflect this dependence and include a reference to Remark 2.1 in [2].
>
> Additionally, we will add a clarification that our choice of $a$ in the decreasing scheme $\varepsilon_k = 1/k^a$ (used in the convergence rate of DRAG in Theorem 2) ensures that the terms involving $K_0$ become negligible as the number of iterations increases. More precisely, they are of the order $o(||\bar{\mathbf{g}}_t - \mathbf{g}^*||^2)$ and therefore do not affect the leading convergence rate.
>
> ### **References**
>
> [1]  Moulines, E., \& Bach, F. (2011). Non-asymptotic analysis of stochastic approximation algorithms for machine learning. Advances in neural information processing systems, 24.
>
> [2] Delalande, A. (2022). Nearly tight convergence bounds for semi-discrete entropic optimal transport.  International Conference on Artificial Intelligence and Statistics (pp. 1619-1642). PMLR.

---

> > ### Comment · Reviewer_nRvS · 2025-08-05
> >
> > I thank the authors for replying to me. I think their suggested improvements will improve the paper, however I will keep my original score.

---

### Decision · Program_Chairs · 2025-09-17

**Decision:**

Accept (poster)

**Comment:**

**Strengths:**

This paper provides the first analysis of a method for optimal transport that anneals the entropic regularization. This is a natural heuristic that is already used in practice [20,34,48,49] but the theoretical analysis of this scheme is open [12,48,52].

The paper is well-written.

Many of the core ideas, specifically showing that a local restricted strong convexity property holds with regularization-independent parameters—seems both original and non-trivial (reviewer XDRF)

The algorithmic design of DRAG is desirable in the context of large-scale learning. Not only it unbiasedly converges to the optimal solution (OT map / OT potential), it is also doing so in an online manner -- using one sample or a minibatch of samples at each iteration. (reviewer Ueg5)


**Weaknesses:**

It was not clear to me from reading the reviews and the paper if the O(1/t) rate was novel.

Experiments were relatively small scale.